# $^{13}$C- and $^{15}$N-labeling of amyloid-β and inhibitory peptides to study their interaction via nanoscale infrared spectroscopy

Suman Paul[1,4], Adéla Jeništová[1], Faraz Vosough [1], Elina Berntsson[1,2], Cecilia Mörman[1,5], Jüri Jarvet [1,3], Astrid Gräslund[1], Sebastian K. T. S. Wärmländer[1] & Andreas Barth [1✉]

Interactions between molecules are fundamental in biology. They occur also between amyloidogenic peptides or proteins that are associated with different amyloid diseases, which makes it important to study the mutual influence of two polypeptides on each other's properties in mixed samples. However, addressing this research question with imaging techniques faces the challenge to distinguish different polypeptides without adding artificial probes for detection. Here, we show that nanoscale infrared spectroscopy in combination with $^{13}$C, $^{15}$N-labeling solves this problem. We studied aggregated amyloid-β peptide (Aβ) and its interaction with an inhibitory peptide (NCAM1-PrP) using scattering-type scanning near-field optical microscopy. Although having similar secondary structure, labeled and unlabeled peptides could be distinguished by comparing optical phase images taken at wavenumbers characteristic for either the labeled or the unlabeled peptide. NCAM1-PrP seems to be able to associate with or to dissolve existing Aβ fibrils because pure Aβ fibrils were not detected after mixing.

[1] Department of Biochemistry and Biophysics, Stockholm University, Stockholm, Sweden. [2] Department of Chemistry and Biotechnology, Tallinn University of Technology, Tallinn, Estonia. [3] National Institute of Chemical Physics and Biophysics, Tallinn, Estonia. [4] Present address: attocube systems AG, Haar, Germany. [5] Present address: Department of Biosciences and Nutrition, Karolinska Institutet, Huddinge, Sweden. ✉email: barth@dbb.su.se

Nanoscale infrared spectroscopy[1,2] is currently revolutionizing the spatial analysis of materials since it combines the morphological information provided by atomic force microscopy (AFM) with the chemical information from infrared spectroscopy. Its spatial resolution beats the diffraction limit by several orders of magnitude and is only limited by the AFM tip radius (~20 nm) and not by the wavelength of the radiation used to interrogate the sample.

There are different technical realizations of nanoscale infrared spectroscopy, which all probe the interaction of a sample with infrared light, but analyze different types of sample responses: photothermal near-field imaging (PTIR or AFM-IR)[3], photo-induced force microscopy (PiFM)[4,5], and scattering-type scanning near-field optical microscopy (s-SNOM)[6–8]. The latter is used in this work and has two operation modes: one for spectrum recording with a broadband infrared laser and one for imaging at a particular wavenumber using a monochromatic laser. These approaches are termed here nano-Fourier transform infrared (FTIR) spectroscopy and s-SNOM infrared imaging, respectively.

All infrared nanospectroscopy techniques are particularly useful to study mixtures, which are of interest in diverse fields of chemistry: e.g., in material science, nanotechnology, biochemistry, and medical chemistry. Such mixtures may be composite materials, self-assembled molecular structures, multi-component colloids, or blends of biomolecules, which are studied for their biological function or for engineering new compounds with designed properties.

Infrared spectroscopy can identify the location of each mixture component and thus reveal its morphology, as long as the chemical structures of the mixture components are sufficiently distinct to generate infrared absorption bands in different regions of the spectrum. However, chemically similar molecules generate similar infrared spectra and are therefore difficult to distinguish. In such cases, it is useful to label one of the mixture components with a stable isotope. The different mass of the labeled component shifts the spectral position of its absorption bands and makes its spectrum distinguishable from that of the unlabeled compound(s). This approach has been used in infrared nanospectroscopy to distinguish synthetic polymers by deuteration[9,10], which generates a large isotope shift and moves the absorption band of the deuterated groups into a spectral region that is largely clear from the absorption of other groups.

Interactions between proteins or polypeptides are important for their biological function and misfunction, which makes it relevant to study protein mixtures. Proteins with different secondary structures can be distinguished by nanoscale infrared spectroscopy because of the different spectral positions of the amide I band of the protein backbone[11]. However, proteins with similar secondary structures have not yet been discriminated in nanoscale images of infrared absorption. The present work demonstrates that this is possible when one of the mixture components is labeled with $^{13}C$- and $^{15}N$-isotopes, which are routinely employed in NMR structure determination[12].

A field of research where a distinction between different peptides with similar secondary structure is relevant is that of amyloidogenic peptides and proteins. They serve important biological functions from bacteria to humans[13] and can be exploited to generate biotechnological products (e.g., silk-based fibers)[14]. But they are most known for their detrimental role in neurodegeneration and other disorders[15]. Bulk[16–24] and nanoscale[25–36] infrared spectroscopy have provided important contributions to elucidate the process of amyloid formation.

Peptides and proteins involved in different amyloid diseases interact with each other, which may provide important links between these diseases[37–42]. Unfortunately, this cross-reactivity is presently underexplored although a better understanding is highly desired. It can be expected to explain the overlap of symptoms that currently makes neurodegenerative diseases difficult to diagnose. This will lead to a better diagnosis and in turn to a more accurate selection of patients for clinical trials and for future treatment with personalized medication.

However, mixtures of amyloidogenic peptides/proteins provoke methodological problems. Their various aggregates may be difficult to distinguish from their morphology alone, in particular when the interactions modify the size and shape of the aggregates. A further problem is that amyloidogenic peptides/proteins are rich in β-sheet structure so their similar secondary structure generates similar infrared spectra, which prevents an identification. This is also true for polypeptides which function as aggregation modulators or inhibitors (antibodies, BRICHOS domain, inhibitory peptides), some of which form amyloid fibrils themselves. Other interacting polypeptides may be induced to form fibrils or to adopt β-sheet structure via interaction with an amyloidogenic peptide/protein, which again makes the morphological and structural properties of the interaction partners difficult to distinguish. For these reasons, important interactions relevant to amyloid diseases are difficult to study with current approaches.

This work focuses on the amyloid-β (Aβ) peptide, which has been ascribed a decisive role in the progression of the most common neurodegenerative disease: Alzheimer's disease (AD)[43–45]. In the disease, the Aβ peptide aggregates to amyloid fibrils that accumulate in plaques in the human brain. These deposits contain predominantly the 42-residue variant of Aβ (Aβ42)[45], although more of the 40-residue variant (Aβ40) is generated in vivo[46].

We use the interaction of Aβ with the inhibitory peptide NCAM1-PrP as an example to demonstrate the value of isotope editing for studying the interaction between polypeptides by nanoscale infrared imaging. NCAM1-PrP was designed by some of us based on the discovery that the N-terminal portion of the nascent prion protein sequence has anti-prion activity[47]. In NCAM1-PrP, the N-terminal prion signal sequence that targets the protein for the secretory pathway was replaced by the equivalent, but shorter, signal sequence from the neural cell adhesion molecule-1 (NCAM1, residues 1–19: MLRTKDLIWTLFFLGTAVS). The C-terminal part of NCAM1-PrP consists of the positively charged hexapeptide KKRPKP, which corresponds to residues 23–28 of the nascent mouse prion protein sequence. NCAM1-PrP aggregates on its own at concentrations of 20 μM and more and forms β-sheet structure in the process. This results in a prominent β-sheet band in the infrared spectrum with a spectral position of 1622 cm$^{-1}$ in D$_2$O[48]. NCAM1-PrP has not only anti-prion activity but inhibits also the aggregation of Aβ in vitro[48–51] by targeting[48] the dominant mechanism for Aβ fibril formation: fibril catalyzed formation of new fibrils (secondary nucleation)[52,53]. It also counteracts the lethal damage inflicted by the Aβ peptide on neuroblastoma cells[50]. The cationic hexapeptide[50] and the signal sequence[47,49] are both important for the anti-amyloid activity and the latter seems to target a relevant cellular location in the neuronal cells[49,50]. These properties of NCAM1-PrP make it a promising starting point for developing a treatment against AD.

This work reports s-SNOM measurements in the infrared spectral range of $^{13}C$, $^{15}N$-labeled Aβ40, and of unlabeled Aβ40, Aβ42, and NCAM1-PrP, either in pure form or in mixtures. We discuss first the nano-FTIR spectra of Aβ40 and its s-SNOM infrared images recorded at two different wavenumbers of which one detects mainly the absorbance of unlabeled peptide and the other mainly the absorbance of labeled peptide. The images of labeled Aβ40 and unlabeled Aβ40 and of their mixture serve to verify the concept of isotope-edited imaging. As a further proof of concept, we analyze images of Aβ42 protofibrils and their mixture

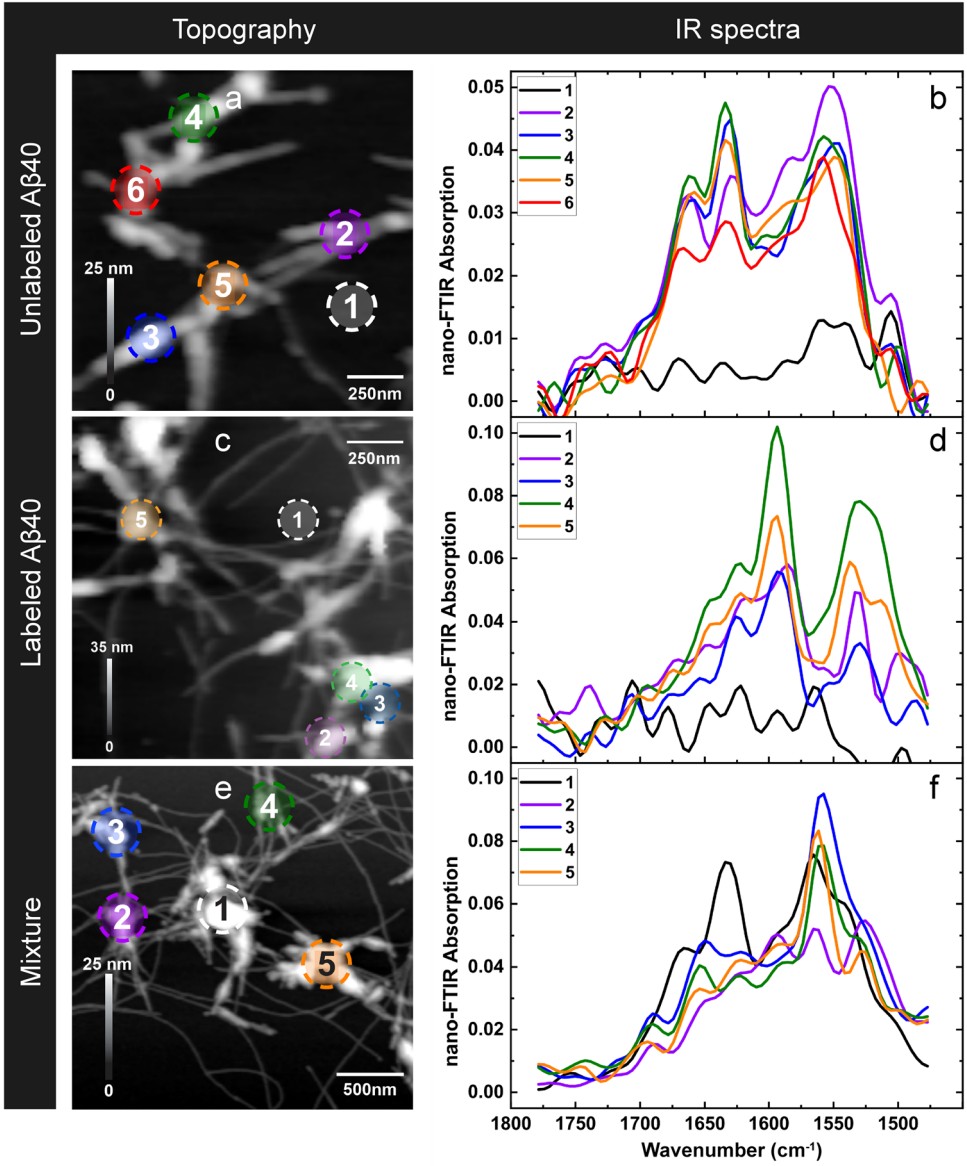

**Fig. 1 Infrared nanospectroscopy of Aβ40 samples.** Aβ40 was prepared from 20 μM solutions at pH 7.4. **a**, **b** unlabeled Aβ40; **c**, **d** $^{13}$C-,$^{15}$N-labeled Aβ40; **e**, **f** 1:1 mixture of labeled and unlabeled Aβ40. **a**, **c**, **e** Height image; **b**, **d**, **f** nano-FTIR spectra in the spectral range of the amide I and amide II absorptions of the polypeptide backbone. A nano-FTIR absorption of 0.1 corresponds to a phase of ~9 degrees.

with labeled Aβ40 fibrils, which demonstrate that the peptide identity determined from the isotope-edited infrared image correlates with the expected aggregate type. Finally, we discuss nano-FTIR spectra and images of NCAM1-PrP fibrils and the images of their mixture with labeled Aβ40 fibrils, which reveal the anti-amyloid effect of the former peptide.

## Results and discussion

**Nano-FTIR spectra of labeled and unlabeled Aβ40.** We first analyze the infrared spectra of two Aβ40 samples: unlabeled Aβ40 and $^{13}$C, $^{15}$N-labeled Aβ40, which are shown in Fig. 1b, d and averaged over several locations in Fig. S1. Also shown in Fig. 1f are spectra of a 1:1 mixture of labeled and unlabeled Aβ40, which will be discussed later. Labeled and unlabeled Aβ samples (20 μM, pH 7.4) were left to aggregate for 60 h at 37 °C and analyzed directly afterward as described in Methods. The spectra were taken at specific points of the samples with a neaSNOM infrared nanospectrometer (Neaspec, Munich, Germany). They show the

nano-FTIR absorption, defined previously[54,55], which is the imaginary part of the normalized scattering coefficient.

The spectra of unlabeled Aβ40 in Fig. 1b clearly demonstrate the presence of the amide I (1700–1600 cm$^{-1}$) and amide II (1600–1500 cm$^{-1}$) bands of polypeptides. They originate from predominantly the CO stretching vibrations (amide I vibration) and the NH bending and CN stretching vibrations (amide II) of the polypeptide backbone[56]. The amide I band position of the unlabeled peptide is at 1632 cm$^{-1}$ (average of the position in five spectra, standard error: 1 cm$^{-1}$), as expected for β-sheet-rich structures[16,57–59].

The amide II band is found at 1553 cm$^{-1}$ (average of the position in 5 spectra, standard error 2 cm$^{-1}$) which is 10–20 cm$^{-1}$ higher than expected for antiparallel β-sheet structures[60,61] but in line with previous Aβ spectra[62–65]. Its position testifies to the formation of parallel β-sheets in the transition from Aβ oligomers to fibrils[63].

The amide II band is relatively strong compared to the amide I band, which is likely due to a polarization effect. The near-field

under the AFM tip is polarized perpendicular to the sample surface. Thus transitions with a transition dipole moment (TDM) that is perpendicular to the surface will absorb strongly, but those with a parallel TDM only weakly. For parallel and antiparallel β-sheets, the amide I vibration that normally absorbs the most is polarized perpendicular to the strand direction in the plane of the sheet[66–68]. Since the strand direction is perpendicular to the fibril axis for amyloid fibrils, the TDM is always parallel to the surface and the vibration absorbs only weakly in nano-FTIR spectra. In contrast, the TDM of the amide II vibration is largely parallel to the strand direction[66,67], which gives rise to considerable absorption whenever the strand direction is not parallel to the sample surface. This is the case because the amyloid fibrils are twisted[69]. Therefore we expect a stronger relative amide II contribution to the nano-FTIR spectra than to solution FTIR spectra of bulk samples. The opposite effect has been observed for the purple membrane[11], which has the α-helical protein bacteriorhodopsin as its main constituent. Here, the amide I band is much stronger than the amide II band because the amide I TDM is perpendicular to the sample surface, whereas the amide II TDM is parallel to the surface.

Spectra of labeled Aβ40 are presented in Fig. 1d. $^{13}$C-labeling shifts the amide I band ~40 cm$^{-1}$ to lower wavenumbers[58,70–72] because the vibration is slowed down by the larger mass of the carbon atom. In contrast, $^{15}$N-labeling has very little influence on the amide I spectrum[70,71]. Thus, $^{13}$C-labeling alone can be expected to work similarly well as $^{13}$C, $^{15}$N-labeling for isotope-edited infrared imaging, and our use of doubly labeled peptides was prompted by their better availability. According to these expectations, the absorption maximum of the amide I vibration of the labeled peptide is observed near 1592 cm$^{-1}$ (average of the position in 4 spectra, standard error: 2 cm$^{-1}$), which corresponds to a downshift of 40 ± 3 cm$^{-1}$. Also the amide II band is affected by the isotopic substitution. Its spectral position for the $^{13}$C, $^{15}$N-labeled peptide is 1533 cm$^{-1}$ (average of the position in 4 spectra, standard error 2 cm$^{-1}$) which corresponds to a downshift of 20 ± 4 cm$^{-1}$. The isotope effect of the amide II vibration is due to the contribution of the C–N stretching vibration. Both $^{13}$C- and $^{15}$N-labeling shifts this band down by 10–15 cm$^{-1}$ [73–76], and the combined effect is a downshift of 25–30 cm$^{-1}$ [74,76,77], which is in line with our results. The isotope shift of the amide II band implies, that it is also useful to discriminate between labeled and unlabeled peptides.

Our spectra of labeled Aβ40 exhibit a reduced amide II absorption relative to the amide I band (Fig. 1d and Fig. S1). This can be tentatively explained by the effects of isotope-labeling, which is known not only to affect the vibrational frequency of a normal mode, but also to alter its internal coordinate contributions—with possible consequences for the direction and magnitude of its TDM. Density functional theory calculations for an amide model compound[78] indicate only small changes in the TDM orientation upon either $^{13}$C- or $^{15}$N-labeling. The magnitude of the amide I transition dipole moment is slightly reduced by $^{13}$C-labeling (4%) but not by $^{15}$N-labeling. However, both isotopes reduce the magnitude of the amide II TDM: $^{13}$C-labeling by 9% and $^{15}$N-labeling by 13%. Assuming additivity of the isotope effects on the amide II TDM magnitude and considering that the integrated absorbance is proportional to the squared TDM magnitude, we estimate a reduction of the $^{13}$C, $^{15}$N-amide II band by 38% relative to the unlabeled amide II band and a corresponding reduction of the amide I band by 9%. Thus, the amide II band is expected to be considerably weaker relative to the amide I band for the labeled peptide than for the unlabeled peptide, which is in line with our experimental spectra.

Figure 1b, d show also control spectra (black lines) taken in regions without the obvious presence of peptides and these spectra have clearly less intensity than those of peptide aggregates. We note that all spectra seem to contain oscillations, which is in contrast to the expected smooth appearance of the protein and the control spectra, and tentatively ascribe the apparent oscillations to the noise. Noise leads to a deviation of the measured data points from the correct spectrum. Since the wavenumber interval between two data points is 16 cm$^{-1}$ at the spectral resolution used (ignoring the data points introduced by zero-filling), the noise will generate minima and maxima in the measured spectrum with a spacing of 16 cm$^{-1}$. This gives the impression of an oscillation with a periodicity of 32 cm$^{-1}$, corresponding to twice the resolution. This is in fact the main apparent oscillation period observed in the data.

**S-SNOM infrared images of labeled and unlabeled Aβ40**. We continue with discussing images of topographical and optical properties of the unlabeled and labeled Aβ40 samples. Please note that the laser power per wavenumber interval is much higher for the quantum cascade laser used for imaging than for the broad-band laser used for taking spectra. Therefore the signal-to-noise ratio is expected to be better for s-SNOM images than for nano-FTIR spectra. Fig. 2 shows AFM and infrared imaging results for unlabeled Aβ40 (panels a–f), $^{13}$C, $^{15}$N-labeled Aβ40 (panels g–l), and a 1:1 mixture of both (panels m-r). Panels a, b, g, h, m, and n illustrate the AFM results, which reveal the presence of many fibrillar structures as well as of amorphous aggregates. The latter was up to 27 nm thick. An analysis of the heights of 43 fibril segments showed that 16% were thinner than 2.0 nm (smallest height: 0.5 nm), 26% between 2.0 and 2.8 nm, and ~20% each had height ranges of 3.4–4.2 nm, 4.4–5.6 nm, and 6.0–10.2 nm. The height range of the fibrils in our study agrees with that of previous AFM studies, which have identified several Aβ40 fibril types with heights between 0.65 to 12 nm[79–82]. Fibrils with heights around 2 nm were found to be composed of 2 filaments, and those with heights of ~4 nm of 4–6 filaments. More than 6 filaments increase the height only little[81]. When compared with electron microscopy images, the AFM-derived widths are larger even in high-resolution AFM measurements that use tips with ~1 nm diameter, while the heights are considerably smaller. The latter may be explained by the arrangement of filament pairs on the substrate[81], the effect of different electrostatic interactions of the tip with the sample and the substrate[83], or the compression of soft biomolecules caused by the interaction with the AFM tip[79,84,85].

The aggregates are also visible in s-SNOM infrared images recorded at a particular wavenumber. Panels c, i, and o show images of the near-field amplitude of the scattered radiation, which is related to the reflectivity of the sample. It is an additional way to visualize the sample, but not used to interrogate its spectral properties. Therefore, we show only the optical amplitude at one of the two wavenumbers and we arbitrarily selected that at 1629 cm$^{-1}$. Most relevant for the present work are the images of the optical phase in panels d, e, j k, p, and q, because the optical phase is an approximation for the imaginary part of the normalized scattering coefficient, which is proportional to the sample absorbance[55]. For the small optical phase values in our work, the optical phase is therefore approximately proportional to the sample absorbance.

We imaged the optical phase at 1629 cm$^{-1}$ and 1587 cm$^{-1}$. For aggregated, unlabeled Aβ, the former wavenumber is close to the absorption maximum of the amide I vibration and the latter close to the absorption minimum between the amide I and amide II absorption bands in our nano-FTIR spectra (Fig. 1a and Fig. S1) and in bulk measurements[86–88]. Therefore the optical phase is expected to be larger at 1629 cm$^{-1}$ than at 1587 cm$^{-1}$ for the unlabeled

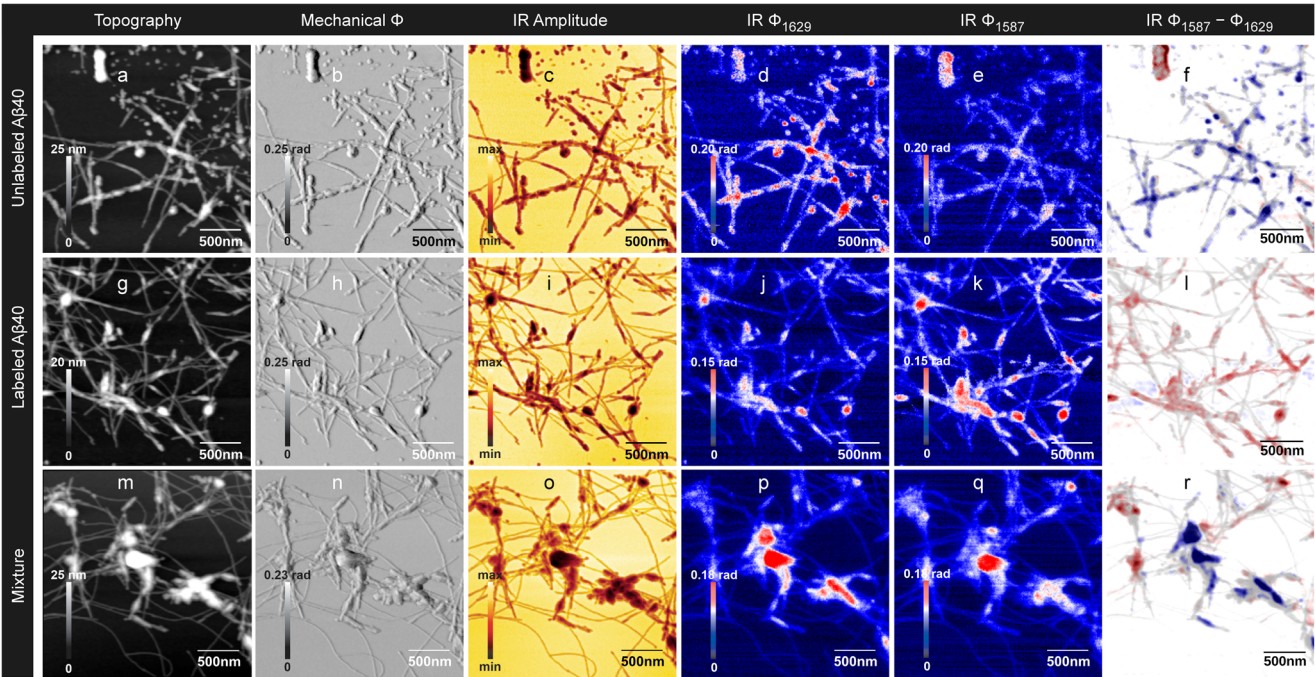

**Fig. 2 Nanoscale infrared imaging of Aβ40 samples. a–f** Unlabeled Aβ40; **g–l** $^{13}$C-,$^{15}$N-labeled Aβ40; **m–r** 1:1 mixture of labeled and unlabeled Aβ40. **a**, **g**, **m** Height image; **b**, **h**, **n** mechanical phase; **c**, **i**, **o** optical amplitude at 1629 cm$^{-1}$; **d**, **j**, **p** optical phase at 1629 cm$^{-1}$; **e**, **k**, **q** optical phase at 1587 cm$^{-1}$; **f**, **l**, **r** difference image of the optical phase at 1587 cm$^{-1}$ minus optical phase at 1629 cm$^{-1}$ overlayed with a height image. Positive values in the difference image are coded in red and indicate more absorption at 1587 cm$^{-1}$ than at 1629 cm$^{-1}$, which is characteristic of the labeled peptide. Negative values are shown in blue and reveal more absorption at 1629 cm$^{-1}$, which is characteristic of the unlabeled peptide. Phase differences close to zero are shown in white. The scale of the phase difference image is symmetrical around zero, i.e., the absolute values of the maximum and minimum scale values are the same. The overlayed semi-transparent height image indicates the height in a grayscale. Gray regions without a blue or red color tone indicate peptide aggregates where the phase difference is close to zero.

peptide. This is indeed the case, when the two optical phase images are compared in Fig. 2d, e. There is non-fibrillar material on the top of these images, which absorbs stronger at 1587 cm$^{-1}$ than at 1629 cm$^{-1}$. We attribute this material to an impurity because it has a well defined shape and is rather isolated from other material. In particular, it is well separated from fibrillar structures. In contrast, amorphous material of similar dimensions is well-integrated into a network of fibrils in all other images (see Figs. 2–4).

For aggregated, $^{13}$C, $^{15}$N-labeled Aβ, the imaging wavenumber of 1587 cm$^{-1}$ is close to its amide I absorption maximum, as shown in Fig. 1d and Fig. S1 and expected from the ~40 cm$^{-1}$ isotope shift. The second imaging wavenumber of 1629 cm$^{-1}$ is in a spectral region where the absorbance declines to higher wavenumbers. Thus, the labeled peptide is expected to have larger optical phase at 1587 cm$^{-1}$ (Fig. 2k) than at 1629 cm$^{-1}$ (Fig. 2j), which is indeed observed. These results indicate that it is possible to distinguish $^{13}$C, $^{15}$N-labeled and unlabeled peptides in nanoscale images of their infrared absorption.

**S-SNOM infrared imaging of a mixture of labeled and unlabeled Aβ40.** We continued to study mixtures of labeled and unlabeled Aβ40 fibrils after separate incubation for 60 h. The results for the 1:1 mixture are shown in Fig. 2m–r. The optical phase at 1629 cm$^{-1}$ is stronger in the central part of the image (Fig. 2p), whereas it is larger at 1587 cm$^{-1}$ (Fig. 2q) in the left and top right regions. This indicates that unlabeled Aβ resides mainly in the central regions of the image, whereas labeled Aβ is located in the left and top right regions. We note that it was difficult to find areas, such as the one shown, where both types of peptides were found in one image. In many cases, the image showed a large predominance of either the labeled or the unlabeled peptide,

in particular when the peptides were mixed without vortexing. We speculate that this is due to the repulsion between extensive networks of negatively charged fibrils, which impedes a homogeneous mixing of labeled and unlabeled peptides. Because of the small area of our images, most of them contained material from either a labeled or an unlabeled network of fibrils. It may be argued against this explanation that the formation of fibrils from Aβ monomers is thermodynamically favorable in spite of the electrostatic repulsion between the Aβ peptides. However, this might either not apply to the association of already formed fibrils or the time between mixing and sample preparation was too short to allow fibril association to proceed to a significant extent.

**Phase difference images and phase line profiles of the Aβ40 samples.** The apparent strength of signals in the phase images depends on the level of the surrounding background. Therefore, a visual inspection of two-phase images recorded at different wavenumbers may lead to erroneous conclusions when the background colors are not carefully matched (which they are in Fig. 2). To avoid any ambiguity, we employed an objective way to compare two-phase images: we calculated difference images where the phase at 1629 cm$^{-1}$ was subtracted from the phase at 1587 cm$^{-1}$. The confidence level in these difference images is analyzed in the SI at the example of a difference image generated from repeated scans at one particular wavenumber (Fig. S3).

The difference images are shown in panels f, l, and r of Fig. 2 as well as in Figs. S4 and S5. Positive values are coded in red and indicate more absorption at 1587 cm$^{-1}$, which is characteristic of the labeled peptide. Negative values are shown in blue and reveal more absorption at 1629 cm$^{-1}$, which is characteristic of the unlabeled peptide. Regions, where the absorption at the two

wavenumbers is similar, are shown in white. Such regions may appear for two reasons: either labeled and unlabeled peptides occur in similar proportions, or none of the peptides occurs there. In order to distinguish between these possibilities, the difference images were overlaid with a semi-transparent height image, where a larger height is shown by a darker shade of gray. In the resulting image, gray areas indicate a mixture of the two peptides, red areas predominantly labeled peptide, blue areas predominantly unlabeled peptides, and white areas no peptides.

The difference images for samples that contained either exclusively unlabeled Aβ40 or labeled Aβ40 served as controls. As expected, they show mainly blue structures when the peptide is unlabeled and mainly red features when the peptide is labeled. There are a few areas with unexpected color in these images, which are mainly outside or at the edges of the structural features and may result from background distortions or a slight misalignment of the two-phase images. Therefore, it is advisable to inspect additionally a plot of the phase along lines that run across structural features of interest as shown in Fig. S4. For the phase profiles of unlabeled Aβ40, the phase at $1629\,\mathrm{cm}^{-1}$ is larger than the phase at $1587\,\mathrm{cm}^{-1}$, whereas the opposite is true for labeled Aβ40. This is expected from the isotope effect on the spectral position of the absorption maximum. The line profiles and the difference images demonstrate the sensitivity of the method to distinguish between labeled and unlabeled structures: even fibrils with a height of only 2 nm show the expected behavior (larger phase at $1587\,\mathrm{cm}^{-1}$ for labeled peptide and larger phase at $1629\,\mathrm{cm}^{-1}$ for unlabeled peptide). This is evident in line profile 1 for unlabeled Aβ40 (Fig. S4d) and line profiles 1 and 2 for labeled Aβ40 (Fig. S4h). An exception is the last fibril in line profile 4 of the unlabeled Aβ40 sample (Fig. S4d), which shows equal phase values at both wavenumbers. This line was deliberately drawn across a region of the difference images with the unexpected color and shows that difference image and line profiles lead to the same conclusions. The reason for the deviation from expectation in this border region of the image is presently unclear, but we note that the two-phase images of this sample were particularly difficult to align with each other because considerably different shifts were required in different parts of the image.

Next, we analyzed whether the phase difference image can discriminate between labeled and unlabeled peptides when both are present in the same image. The difference image of the mixture is shown in Fig. 2r, where clear regions with predominantly labeled and predominantly unlabeled peptides can be distinguished (red and blue areas, respectively). This is also evident in the corresponding phase profiles shown in Fig. S5d. Thin fibrils show a more equal absorption at 1587 and $1629\,\mathrm{cm}^{-1}$ than the control samples with pure material, which seems to indicate some mixing.

**Nano-FTIR spectra of a mixture of labeled and unlabeled Aβ40.** The interpretation of the difference image is supported by the nano-FTIR spectra of the mixed sample shown in Fig. 1f. Spectrum 1 was recorded from a region with predominantly unlabeled peptide according to the difference image in Fig. 2r. In line with this, its amide I band position is near $1630\,\mathrm{cm}^{-1}$ and its amide II position is high in Fig. 1f. In contrast, spectrum 2 was taken from a region with predominantly labeled peptide and accordingly its amide I maximum is near $1590\,\mathrm{cm}^{-1}$ in Fig. 1f and it has a strong low wavenumber amide II band near $1530\,\mathrm{cm}^{-1}$. The other spectra originate from regions with mixed peptide composition and therefore exhibit a flatter absorption in the $1630$–$1590\,\mathrm{cm}^{-1}$ region: spectrum 3 stems from just above the red area in the difference image, spectrum 4 was taken right of the two small blue regions in the difference image and spectrum 5 was recorded close to a narrow blue ridge. All three spectra exhibit a high wavenumber of the amide II band near $1560\,\mathrm{cm}^{-1}$ indicating the predominance of unlabeled peptide. A shoulder near $1530\,\mathrm{cm}^{-1}$ indicates the additional presence of labeled peptide at least in spectra 4 and 5.

**Mixture of labeled Aβ40 fibrils and of unlabeled Aβ42 protofibrils.** We next turned to a mixture of different types of aggregates: fibrils of labeled Aβ40 and protofibrils of unlabeled Aβ42 prepared by gel filtration (see Methods). Images of the pure Aβ42 sample are shown in Fig. 3a–f and of the mixture in Fig. 3g–l. For the pure Aβ42 sample, short fibrillar aggregates are visible in the mechanical and optical images. As expected for an

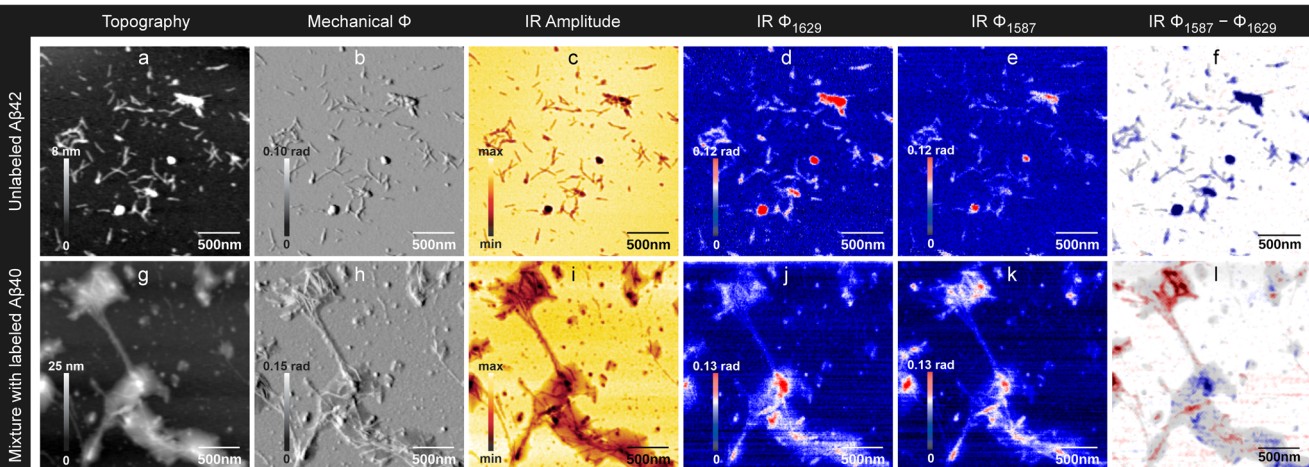

**Fig. 3 Nanoscale infrared imaging of Aβ42 and of a 1:1 mixture of Aβ40 and Aβ42.** Aβ40 and Aβ42 were prepared from 20 µM solutions at pH 7.4–7.5. **a–f** unlabeled Aβ42; **g–l** 1:1 mixture of unlabeled Aβ42 and labeled Aβ40. **a, g** Height image; **b, h** mechanical phase; **c, i** optical amplitude at $1629\,\mathrm{cm}^{-1}$; **d, j** optical phase at $1629\,\mathrm{cm}^{-1}$; **e, k** optical phase at $1587\,\mathrm{cm}^{-1}$, **f, l** optical phase at $1587\,\mathrm{cm}^{-1}$ minus optical phase at $1629\,\mathrm{cm}^{-1}$ overlayed with a height image. Positive values in the difference image are coded in red and indicate more absorption at $1587\,\mathrm{cm}^{-1}$ than at $1629\,\mathrm{cm}^{-1}$, which is characteristic of the labeled peptide. Negative values are shown in blue and reveal more absorption at $1629\,\mathrm{cm}^{-1}$, which is characteristic of the unlabeled peptide. Phase differences close to zero are shown in white. The scale of the phase difference image is symmetrical around zero, i.e. the absolute values of the maximum and minimum scale values are the same. The overlayed semi-transparent height image indicates the height in a gray scale. Gray regions without a blue or red color tone indicate peptide aggregates where the phase difference is close to zero.

unlabeled peptide, the phase signal at 1629 cm$^{-1}$ is stronger than that at 1587 cm$^{-1}$. This is also confirmed by the difference image which shows predominantly the blue color for the unlabeled peptide. Accordingly, most line profiles for this sample in Fig. S4l show a larger signal at 1629 cm$^{-1}$ (blue) than at 1587 cm$^{-1}$ (red) but there are a few exceptions where the signal is equal. These are generally aggregates with weak signals.

The mixed sample shows an interesting morphology: a core of dense aggregates covered or surrounded by a blanket of unstructured material. The fibrillar nature of the core aggregates is best appreciated in the mechanical phase image. According to the difference image, the fibrillar material is mainly composed of labeled Aβ40 (red) and the surrounding unstructured material is mainly a mixture of both peptides. This is particularly evident in the top left corner. It can also be seen in the line profiles in Fig. S5h, which reveal stronger signals at 1587 cm$^{-1}$ when passing through fibrils (e.g., line 3, first two peaks of line 4, central, sharp peak of line 5, line 6, first peak of line 7, line 10). In contrast, small spherical aggregates tend to be of mixed composition or to consist mainly of Aβ42 (second and third peak of line 7, line 8, line 12, line 13).

**Nano-FTIR spectrum and s-SNOM infrared imaging of NCAM1-PrP**. The nano-FTIR spectrum of NCAM1-PrP is shown in Fig. S1. It exhibits amide I and II maxima at 1640 and 1533 cm$^{-1}$, respectively, which both indicate β-sheet structure. The low amide II band position indicates additionally an anti-parallel arrangement of the β-strands in contrast to the parallel orientation in Aβ40 fibrils discussed above. Antiparallel β-sheets of the NCAM1-PrP fibrils are also indicated by their FTIR spectrum (dried from H$_2$O solution, measured by attenuated total reflection, included in the data set published on figshare: https://doi.org/10.17045/sthlmuni.23609580.v1), which exhibits typical antiparallel β-sheet features: a distinct high wavenumber amide I band at 1696 cm$^{-1}$, a main amide I band at 1627 cm$^{-1}$, and an amide II maximum at 1526 cm$^{-1}$. While the amide II band positions are close in nano-FTIR and FTIR spectra, the amide I band positions are significantly different, which we ascribe to the polarization effect discussed above. A strong low wavenumber β-sheet signal in the amide I range would require the β-sheets to be oriented perpendicular to the substrate surface with strands running parallel to the surface. The former is not possible given the thinness of the fibrils (see below).

In spite of the differences between the nano-FTIR spectrum and the FTIR spectrum in the amide I range, the nano-FTIR spectrum demonstrates that the signal at 1629 cm$^{-1}$ is stronger than that at 1587 cm$^{-1}$ and therefore that the wavenumbers used for imaging in the previous sections are also suitable for studying mixtures of unlabeled NCAM1-PrP and labeled Aβ40.

NCAM1-PrP (20 μM, pH 7.4) forms fibrils on its own[89], as also shown in Fig. 4a–f. Individual fibrils have a height of 2.7 nm (average from 40 fibrils, standard deviation 0.2 nm) while the thickest aggregates are ~5 nm high. As expected, the optical phase and difference images indicate that the phase at 1629 cm$^{-1}$ is larger than at 1587 cm$^{-1}$. This is confirmed by the line profiles in Fig. S4p where even fibrils that appear gray in the difference image are clearly identified to be unlabeled (e.g., lines 3 and 5).

**Interaction of Aβ40 with NCAM1-PrP**. Finally, we investigated the interaction of the anti-amyloid peptide NCAM1-PrP with Aβ40. The results of mixing fibrillar NCAM1-PrP and labeled, fibrillar Aβ40 are shown in Fig. 4g–l and in Fig. S5i–p. The latter figure shows also an additional sample area compared to Fig. 4. If the two peptides were not interacting, we would expect aggregates with either a stronger signal at 1629 cm$^{-1}$ (for NCAM1-PrP) or

at 1587 cm$^{-1}$ (for Aβ40). Since the pure Aβ40 samples were rich in fibrils, we would expect also pure Aβ40 fibrils in the mixed sample, i.e. fibrils with a stronger signal at 1587 cm$^{-1}$. However, neither the difference images (Fig. 4l, Fig. S5i, m) nor the line profiles (Fig. S5l, p) indicate pure Aβ40 fibrils: there are no red fibrils in the difference images of the NCAM1-PrP+Aβ40 sample that would indicate stronger absorption at 1587 cm$^{-1}$ than at 1629 cm$^{-1}$ and none of the fibrils examined by the line profiles (Fig. S5l, p) exhibits a stronger absorption at 1587 cm$^{-1}$ (red curves) than at 1629 cm$^{-1}$ (blue curves) as would be expected for labeled Aβ40. This is in contrast to the pure, labeled Aβ40 sample, were all examined fibrils display this behavior (Fig. S4h). It is also in contrast to the mixture of labeled and unlabeled Aβ40 and the Aβ42 + Aβ40 mixture (Figs. 2r, 3l, S5d, h), where fibrils with stronger signal at 1587 cm$^{-1}$ can be observed (see Fig. S5d, e.g., lines 3, 4, and 5 and Fig S5h, lines 4, 6, 7, and 10). This demonstrates that labeled fibrils can be detected in mixtures of labeled and unlabeled peptides. For the NCAM1-PrP +Aβ40 sample we conclude that there is no indication of pure Aβ40 fibrils in the presence of NCAM1-PrP. Regions that can be associated with Aβ40 (red areas) in the difference images stem instead from amorphous aggregates (Fig. 4l and Fig. S5i, m).

The fibrils of the NCAM1-PrP+Aβ40 sample show either equal absorption at 1587 and 1629 cm$^{-1}$ in the difference images (gray), characteristic for a mixed composition, or stronger absorption at 1629 cm$^{-1}$ (blue), characteristic for the unlabeled peptide NCAM1-PrP. Similarly, in the line profiles (Fig. S5l, p), the examined fibrils have either similar absorption at 1587 cm$^{-1}$ (red curves) and at 1629 cm$^{-1}$ (blue curves) or stronger absorption at 1629 cm$^{-1}$ (blue curves). The fibrils in our images are therefore either composed of both peptides or predominately of NCAM1-PrP.

Since we did not observe pure Aβ40 fibrils, which were present before mixing, we conclude that mixing with NCAM1-PrP either transformed them to amorphous aggregates or that they became a constituent of fibril co-assemblies. This is not due to the mixing process itself, as labeled Aβ40 fibrils were observed in the mixture of labeled and unlabeled Aβ40 and in the mixture of labeled Aβ40 and unlabeled Aβ42. Thus, we propose that the observed effects are due to the action of NCAM1-PrP, which is associated with or dissolved existing Aβ40 fibrils. It remains to be revealed, which of the NCAM1-PrP species—monomers, oligomers, or fibrils—is responsible for the effect.

Our interpretation is in line with previous findings. NCAM1-PrP was found to inhibit secondary nucleation of Aβ in a similar way as BRICHOS[48], which is known to coat fibrils and thus prevent the surface-catalyzed formation of new fibrils[90]. The analogous action of NCAM1-PrP thus suggests that it can bind to Aβ fibrils and this will be facilitated by the opposite charges of the two peptides. The binding of NCAM1-PrP to Aβ fibrils is supported by our observation of fibrils with similar absorption at 1587 cm$^{-1}$ and at 1629 cm$^{-1}$ indicating that they consist of both Aβ and NCAM1-PrP. Such heterogeneous fibrils are conceivable, since fibrils formed from two different peptides have been discovered previously[91–93].

We observed amorphous Aβ40 aggregates but no pure Aβ40 fibrils after the addition of NCAM1-PrP although the Aβ sample contained fibrils. Also in the interaction with monomeric Aβ, NCAM1-PrP prevents the formation of amyloid fibrils and produces amorphous aggregates[48,50]. A second finding in this context is that NCAM1-PrP promotes the aggregation of the protein S100A9 by destabilizing the native structure and creating a misfolded state[89]. It has therefore been suggested that NCAM1-PrP unfolds or dissolves structured proteins by interaction with amyloidogenic sequences[48], which is in line with our proposal that NCAM1-PrP is able to destabilize Aβ fibrils.

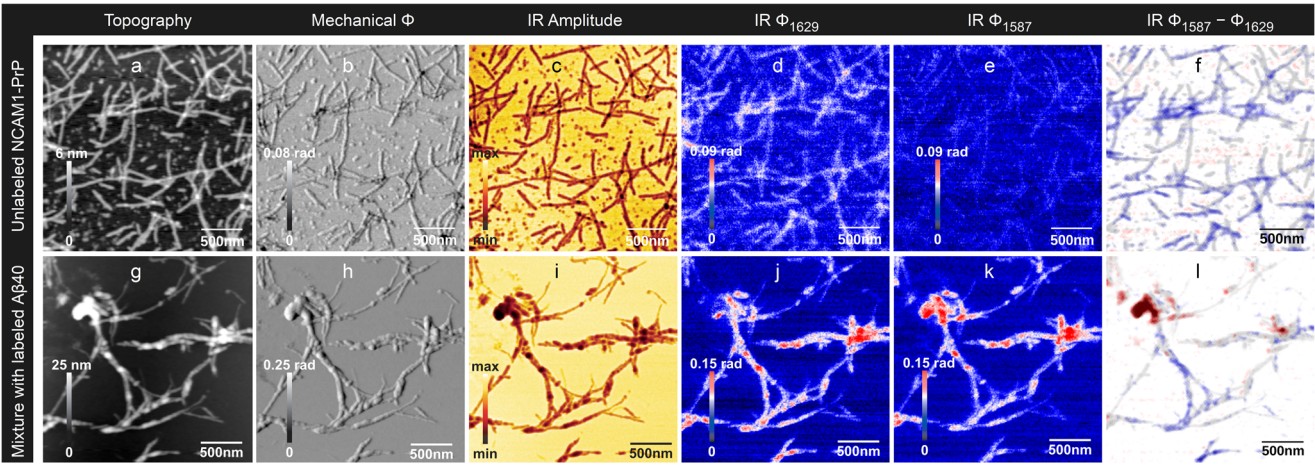

**Fig. 4 Nanoscale infrared imaging of NCAM1-PrP and of a 1:1 mixture of NCAM1-PrP and Aβ40.** NCAM1-PrP and Aβ40 were prepared from 20 µM solutions at pH 7.4. **a–f** unlabeled NCAM1-PrP; **g–l** 1:1 mixture of unlabeled NCAM1-PrP and labeled Aβ40. **a, g** Height image; **b, h** mechanical phase; **c, i** optical amplitude at 1629 cm$^{-1}$; **d, j** optical phase at 1629 cm$^{-1}$; **e, k** optical phase at 1587 cm$^{-1}$; **f, l** optical phase at 1587 cm$^{-1}$ minus optical phase at 1629 cm$^{-1}$ overlayed with a height image. Positive values in the difference image are coded in red and indicate more absorption at 1587 cm$^{-1}$ than at 1629 cm$^{-1}$, which is characteristic of the labeled peptide. Negative values are shown in blue and reveal more absorption at 1629 cm$^{-1}$, which is characteristic of the unlabeled peptide. Phase differences close to zero are shown in white. The scale of the phase difference image is symmetrical around zero, i.e., the absolute values of the maximum and minimum scale values are the same. The overlayed semi-transparent height image indicates the height in a gray scale. Gray regions without a blue or red color tone indicate peptide aggregates where the phase difference is close to zero.

## Conclusion

Our work adds isotope-editing of polypeptides to the toolbox of nanoscale infrared imaging. While we demonstrated its feasibility for s-SNOM infrared imaging, the concept is also applicable to other nanoscale vibrational spectroscopy methods, like PTIR (AFM-IR), PiFM, and tip-enhanced Raman spectroscopy. Due to the small isotope shift of ~40 cm$^{-1}$ and the absence of a clear spectral window for the amide I band of $^{13}$C- or $^{13}$C, $^{15}$N-labeled polypeptides, the approach is challenging to implement. Nevertheless, this work demonstrates that the locations of $^{13}$C, $^{15}$N-labeled, and of unlabeled peptides can be distinguished from their amide I absorption. Also, the amide II band exhibits an isotope shift, which is exploitable for this purpose. As a consequence, the individual morphologies of labeled and unlabeled peptide aggregates can be assessed. Even fibrils as thin as 2 nm were generally classified as either labeled or unlabeled. We obtained also evidence for the mixing of two different polypeptides, which might occur for two reasons: (i) tight and specific molecular interactions between the mixture components, or (ii) a random co-localization of weakly or non-interacting components— besides or on top of each other. These two possibilities cannot be distinguished for large aggregates without a detailed analysis of the spectrum. On the other hand, mixing within small aggregates is likely due to molecular interactions but its unequivocal detection is currently limited by the signal-to-noise ratio. Ongoing technological developments are expected to push the detection limit to even smaller aggregates or molecules.

In future applications, full-range spectra at locations of interest identified by the isotope-edited images will be a valuable complement to the morphology information of the individual interaction partners. While it even would be desirable to have full spectral information for the entire area of nanoscale images (hyperspectral imaging), monitoring at a few selected wavenumbers dramatically speeds up the recording of the images and thus minimizes the risk of tip contamination or tip damage during the experiment. It will enable researchers to scan larger sample areas than with hyperspectral imaging leading to a more representative sampling of the sample properties.

Such imaging of spectral properties at particular wavenumbers or wavelengths has been proven valuable in infrared and Raman microscopy to map particular functional groups and in fluorescence microscopy to map one or several specific molecules. Similar to fluorescence microscopy, the present approach can identify different polypeptides, but has two advantages: (i) it beats the diffraction limit and enables mapping with 20 nm spatial resolution, and (ii) it avoids the need to attach a bulky label for monitoring, which might affect the chemical properties and modify the interactions to be studied.

For very many applications, isotope labeling does not present an obstacle because it can be achieved either by expression of recombinant, labeled proteins or by chemical synthesis of labeled peptides. Studying the interaction of one particular polypeptide— e.g., Aβ—with several other partners can be done in an economic way as it requires only labeling the common polypeptide of interest (Aβ), while each of the other interaction partners may remain unlabeled in binary mixtures. The approach can be extended to more than two interacting polypeptides by imaging one labeled interaction partner at a time in a series of multi-component mixtures.

Thus we expect that isotope-edition of nanoscale infrared images will be useful for studying interacting polypeptides in in vitro studies that serve to understand the natural complexity of biological processes. It will be particularly beneficial for amyloidogenic peptides/proteins where such interactions have medical relevance.

## Methods

**Materials.** NCAM1$_{1–19}$-mPrP$_{23–28}$ (NCAM1-PrP) with the sequence MLRTKDLIWTLFFLGTAVSKKRPKP-NH2 was synthesized by polyPeptide Group, Strasbourg, France. Unlabeled and uniformly $^{13}$C, $^{15}$N-labeled recombinant Aβ40 was obtained from AlexoTech AB (Umeå, Sweden) and recombinant Aβ42 from rPeptide (Watkinsville, USA).

**Peptide preparation.** The lyophilized peptides were stored at −80 °C. Immediately before each experiment, Aβ40 was dissolved at 200 µM in 10 mM NaOH, sonicated in ice bath for 4–5 min to dissolve pre-formed aggregates, and diluted to 20 µM in 10 mM sodium phosphate buffer (pH 7.4). NCAM1-PrP was dissolved on ice in

Milli-Q water at 100 µM and then diluted to 20 µM in the same phosphate buffer. Aβ40 and NCAM1-PrP were distributed equally (50 µL) in multiple Eppendorf tubes and separately incubated for 60 h at 37 °C on a shaker at 300 rpm. Thus, three sets of samples were obtained (1) Aβ40, (2) labeled Aβ40, and (3) NCAM1-PrP. Immediately after the incubation, 1 µL of each sample was used for infrared nanospectroscopy, and from the remaining samples the following 1:1 mixtures were obtained: (4) Aβ40 and labeled Aβ40, (5) NCAM1-PrP and labeled Aβ40. The Aβ40 mixture was vortexed for a few seconds in order to facilitate the observation of both components in a single nanoscale image.

To prepare Aβ42 protofibrils, 1 mg freeze-dried Aβ42 was dissolved in 6 M GuHCl (pH 7.2) at 2 mg/mL and applied on a column for size exclusion chromatography (Superdex 75/300 GL, GE Healthcare, USA). The running buffer was 10 mM sodium phosphate buffer (pH 7.5). The fractions were eluted at room temperature and immediately placed in an ice bath. The fraction of aggregated Aβ42 (protofibrils corresponding to ~50 µM monomer concentration) from the peak prior to the monomeric peak was stored at 5 °C. This protofibril sample is soluble and has Thioflavin T activity. Fluorescence correlation spectroscopy measurements showed that the size distribution was not affected by the storage. After thawing, Aβ42 protofibrils were used directly for the infrared nanospectroscopy experiments either alone or in a 1:1 mixture with labeled Aβ40 fibrils, prepared as described above.

**Sample preparation for infrared nanospectroscopy**. A 1 µL droplet of peptide solution was placed on a fresh silicon wafer (Siegert Wafer GmbH, Germany) and left to dry for 2 min. Next, 10 µL of Milli-Q water was carefully added to the semi-dried sample droplet and soaked away immediately with a lint-free wipe, in order to remove excess salts in a mild manner. The wafer was left to dry in a covered container to protect it from dust, and the infrared nanospectroscopy experiments were performed on the same day.

**Nanoscale imaging of infrared absorption**. We used a neaSNOM instrument (Neaspec/attocube, Germany) in the s-SNOM mode to record nanoscale images of the infrared absorption at two different wavenumbers. They were acquired by measuring the optical phase at the second harmonic of the tip oscillation frequency on 2.5 µm × 2.5 µm scan-areas (200 × 200-pixel) under optimal scan-speed (9.8 ms/pixel) using Pt/Ir-coated monolithic ARROW-NCPt Si tips (NanoAndMore GmbH, Germany) with tip radius <10 nm (tapping frequency ~275 kHz, tapping amplitude 58–62 nm). The laser power was kept in the 0.47–0.50 mW range. This low power does not evaporate or destroy the sample. The laser wavenumber was either 1629 cm$^{-1}$ or 1587 cm$^{-1}$, which is close to either the absorption maxima of unlabeled or of $^{13}$C, $^{15}$N-labeled peptide aggregates, respectively. In addition, both wavenumbers are located at minima of the water vapor absorption spectrum.

The phase images were processed in Gwyddion: Flatten base was used to correct for background distortions and the minimum value was set to zero. The color range of the images was limited to a subset of the phase values, but was the same for images at the two wavenumbers of the same sample. The minimum value of the color range was carefully adjusted so that the background color impression was the same in the images taken at the two wavenumbers.

**Calculation of difference images**. Images of the difference between the optical phase at 1587 cm$^{-1}$ and at 1629 cm$^{-1}$ were obtained with Gwyddion as described in the following. The two images at the two wavenumbers showed slightly different areas. Therefore, the phase images were cropped with the intention to create images that showed the same area. Then the phase values were subtracted (phase at 1587 cm$^{-1}$ minus phase at 1629 cm$^{-1}$), and several crops were judged by inspection of the resulting difference image. When the two cropped images were misaligned, this resulted in features where positive and negative values ran parallel as shown in Fig. S2. The missing of such parallel features was taken as the criterion for a correct alignment of the two cropped images. At a later stage, the alignment of the two-phase images was done with a spreadsheet application using LibreOffice Calc. Some images had different relative shifts in different parts of the image and therefore needed to be cropped differently in different regions for best results. These different crops were then combined to the final difference image. In other cases, the best results were obtained by averaging the results from different crops. The final difference spectrum was further processed in Gwyddion, the rows were aligned if necessary, the resulting phase differences averaged over 4 pixels, and the most abundant pixel value set to zero. Finally, a semi-transparent image of the height of the sample was overlayed with the difference image using LibreOffice Draw. The figures were arranged in QuarkXPress 2018 and LibreOffice Draw.

**Nano-FTIR spectroscopy**. We used s-SNOM with the same neaSNOM instrument for imaging also for spectrum recording. First, low-resolution images of the topography and the mechanical and optical properties were recorded in order to find a location of interest. Then a spectrum was recorded and another set of images recorded to make sure that the sample had not moved relative to the tip during spectrum recording. Finally, a larger, high-resolution image was recorded and the locations of spectrum recorded annotated on this image. This approach reduced the risk of tip damage during the experiment. Generally, we used the same ARROW-NCPt Si tip as for imaging but for the spectra of unlabeled Aβ40, we used

nano-FTIR tips (Neaspec, Germany, tip radius ~30 nm, tapping frequency 230 kHz, tapping amplitude 69 nm). The latter approach was used initially. It had the disadvantage that imaging and spectrum recording could not be performed with the same tip because the nano-FTIR tips were blunter. Therefore, also imaging and spectrum recording could not be performed on the same sample area because of difficulties to return to the same sample area after tip exchange. The spectra were recorded at the second harmonic of the tip oscillation frequency using a broadband infrared laser (1300–2150 cm$^{-1}$, 0.7–0.8 mW), 16 cm$^{-1}$ spectral resolution, a zero-filling factor of 4, 10 ms integration time for each of the 2048 interferogram data points, and averaging 5 or 15 interferograms with the nano-FTIR tips or the ARRPW-NCPt tips, respectively. Recording of one spectrum took thus ~100 s or ~5 min. The reference spectrum was recorded on pure Si. Phase tilt and phase offset were corrected and the nano-FTIR absorption[54,55] calculated with the nea-PLOT software (Neaspec, Germany).

## Data availability

Raw and processed data are available on figshare at https://doi.org/10.17045/sthlmuni.23609580.v1.

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

## Acknowledgements

This work was enabled by grants from Olle Engkvists stiftelse for the infrared nanospectrometer and for postdoc fellowships to Dr. Suman Paul and Adéla Jeništová. We are grateful to Dr. Adrian Cernescu (Neaspec-Attocube Systems, Munich, Germany) for invaluable support and to Niels C. Barth for speeding up data evaluation through the creation of specialized software.

## Author contributions

S.P.—formal analysis, investigation, methodology, project administration, visualization, writing–original draft, writing–review & editing. A.J.—formal analysis, investigation, methodology, visualization, writing–review & editing. F.V.—investigation. E.B.—investigation, methodology, resources, writing–review & editing. C.M.—investigation, methodology, resources, writing–review & editing. J.J.—conceptualization, formal analysis, supervision, writing–review & editing. A.G.—conceptualization, funding acquisition, resources, writing–review & editing. SKTSW—conceptualization, supervision, writing–review & editing. A.B.—conceptualization, formal analysis, funding acquisition, methodology, project administration, supervision, visualization, writing–original draft, writing–review & editing.

## Funding

## Competing interests

The authors declare no competing interests.
