## [Peer Review File · Communications Chemistry]

Reviewers' comments:

Reviewer #1 (Remarks to the Author):

Manuscript untitled: « ^{13}C -isotope-editing of nanoscale infrared images reveals the action of an inhibitory peptide against amyloid- β aggregation »

The article presents a new methodology to differentiate 2 peptides with similar secondary structures. For that purpose, the authors have used an already well-known near-field technique (s-snom) and C^{13} labelling peptides (widely used in classical IR spectroscopy).

It's an interesting approach, challenging to implement. Unfortunately, the work is not completed in its current form: the methodology is not strong enough. I have a lot of questions and comments but I will only send the most important ones.

1. The introduction should be reinforced, directing it more sincerely towards methodology, especially if this article is to be published in a journal like comm.chem. From my point of view, if the authors insist so much (especially in the title) on the application in biology, it would seem to me more judicious to publish this article in a journal related to biophysics... maybe? But again, not as it is.

The authors need to argue more:

2. Why did they choose this NCAM1-Prp hybrid peptide for interactions with A β 40-42? Even if it's clearly described in their previous paper, we need a short reminder. What puzzles me more is the following: some research groups have shown that the species bound to PrPC are predominantly small oligomers (not only fibrils). Would not it be the same for the hybrid peptide: as the authors have the technology to study molecular interaction at the fibrillar level why not explore this aspect?

3. Furthermore, PrP 23–27 region is located within the part of the prion protein that is natively unstructured. If we look at their previous paper (ref 39), compared to A β 42, the IR absorption spectrum of this peptide does not exhibit a high absorption at 1629 cm^{-1} but a broad band centred around 1645-1640 cm^{-1} . So why the authors only acquired the 1629 cm^{-1} IR map? How do the authors justify the lack of spectral data about this NCAM1-Prp hybrid (I mean in the present paper)? This point is crucial for the consistency of the paper. It's a serious methodological issue.

4. Then, spectral measurements were limited to bundles of fibres whereas we are waiting for measurements on individual fibres. Is there a reason? It's almost the same problem for imaging. The interpretation of the signal is more or less clear for bundles but not for singles fibres. The images should clarify this point but as there are a lot of bundles it's impossible to conclude even when looking carefully at supplementary figures.

5. From the spectra it seems that there are a lot of variabilities. How do the authors explain this observation? Why do we observe so many oscillations on spectra 1 (reference)? Why is the amide II band more intense than the amide I band for the unlabeled A β but not for the labelled A β ? Still, we observe a lot of oscillations on the IR spectra, why?

6. What kind of data processing was done on the spectra? It would be useful to have access to the raw spectra.

7. Figure 2, 3, 4: Why the authors do not show the optical amplitude at 1587 cm^{-1} ?

8. Figure 4 top panel: the hybrid peptide seems to exhibit a low signal at 1629 cm⁻¹. Why the signal is higher when it's mixed with the labelled A β 40?
9. In the conclusion, the authors claim that "These observations indicate that NCAM1-PrP is able to coat or to dissolve existing A β 40 fibrils": I don't understand how the authors arrive at this conclusion just by looking at figure 4 and S4.
10. In the SI: the part related to purple membrane is completely irrelevant. The authors should remove it.

Reviewer #2 (Remarks to the Author):

The manuscript by Paul et al. entitled "13C-isotope-editing of nanoscale infrared images reveals the action of an inhibitory peptide against amyloid- β aggregation" described for the first time that 13C labelled proteins or peptides (here on AB fibrils) can be observed and distinguished from the same non-labelled proteins by nano-IR. This is of importance when comparing the same protein or proteins with similar secondary structure. Furthermore, they showed that such labelled spectral features can be observed on sample as thin as 2 nm. Finally, they use their method to study the interaction of the inhibitory NCAM1-Prp on AB aggregation. This is an important addition to both the field of nano-IR and amyloid study.

On page 4, line 10: the authors gave the minimal (2nm) and maximal (12 nm) thickness of the Ab40 fibrils. It would be interesting to know what the average size of the fibrils is. Does it correspond to regular amyloid fibrils (7-10 nm)?

On page 4, lines 21-22: "There is non-fibrillar material on the top of these images, which absorbs stronger at 1587 cm⁻¹ than at 1629 cm⁻¹ and which we attribute to an impurity because of its different shape." In almost all the images, there is high absorbance at 1587 cm⁻¹ in globular shape entities. How to distinguish these from the potential impurity mentioned before? Or the globulomers (potentially oligomers) contains more labelled peptides than fibrils. This is never mentioned or explained.

On page 5, lines 4-5: "In many cases, the image showed a large predominance of either the labeled or the unlabeled peptide". Is there an explanation for this, except for the absence of vortexing?

On page 5, lines 13-15: the authors indicated the color code to understand the difference spectral images (last column of fig 2, 3 and 4 and supplementary figures), it would be clearer to indicate these in the legend of fig 2, the first time the authors showed these images.

On page 5, line 25: "Therefore, it is advisable to inspect additionally a plot of the phase along lines that run across structural features of interest as shown in Fig. S3. » When looking at fig. S3 and S4, spectral plots with Ab42 (and NCAM1-PrP in S3) displayed narrower bands than with Ab40, is there an explanation for this?

In Fig. 1, corresponding to the explanation on page 6 lines 14-18, it is not clear why the spectral features

of the labelled peptide are not present at all in the bottom row. Therefore, how was it possible to calculate the 1587 minus 1629 cm^{-1} images if in spectra from a mixture of labelled and non-labelled peptides the band at 1587 cm^{-1} is absent?

On page 7, the explanation of the role of NCAM1-PrP is difficult to follow. First, the authors stated that none of the mixed fibrils exhibits a stronger absorption at 1587 cm^{-1} . Thus, the formed fibrils should be composed of NCAM1-Prp and not labelled Ab40. But later the authors wrote “the mixed composition of the fibrils in our Ab:NCAM1-Prp images ...”. For us there is no mixed composition of fibrils, once more all the labelled peptides seem to be in globular shape features.

In the legend of fig 4, point E: it is not “optical phase at 1587 cm^{-1} ” but 1629 cm^{-1}

Reviewer #3 (Remarks to the Author):

In “ ^{13}C -isotope-editing of nanoscale infrared images reveals the action of an inhibitory peptide against amyloid- β aggregation” Paul et al. used infrared (IR) scattering-type near-field optical microscopy (s-SNOM) to map the distribution of natural and ^{13}C -labelled amyloid- β peptides deposited on a substrate. s-SNOM is one of few of AFM-based nanoscale IR spectroscopy measurement paradigms which have been widely used for material characterization in the recent literature. Analysis of the conformational arrangement of such amyloids at the nanoscale is important because amyloids aggregates are linked to Alzheimer’s and other diseases and because such aggregates are typically both morphologically and conformationally heterogeneous at the nanoscale. Therefore, nanoscale conformational analysis of such aggregates is being studied extensively.

Although selective isotopic labelling is a widely used strategy in IR spectroscopy to distinguish components within mixtures of otherwise spectrally similar molecules and has been used previously in nanoscale IR applications (see for example ref #33 and *Analyst*, 2018, 143, 3808–3813). I believe this is the first time it has been employed to study peptides at the nanoscale. I should be noted that isotopic labelling is not universally useful, and it is rarely used in samples composed by initially unknown species but may be effective in selected applications, often where the average composition of the sample is known a priori.

The authors propose using difference s-SNOM phase images between two wavelengths (1632 cm^{-1} , characteristic of unlabeled β -sheets and 1587 cm^{-1} , characteristic of ^{13}C β -sheets) to locate the positions of labeled and unlabeled peptides mixed samples. I believe that this type of data analysis has not been reported previously for s-SNOM images but is quite similar to image ratios used for semiquantitative analysis in AFM-IR (another AFM-based nanoscale spectroscopy method that uses the AFM tip to transduce the sample thermal expansion). See for example *Analyst*, 2016, 141, 4848–4854. As the authors point out, such approach has limitations due to image drift that can even vary during a single image scan.

The manuscript is clearly written and while I appreciate the analytical effort put forward by the authors, unluckily I don't find the analysis very useful nor particularly reliable, due to the limitations highlighted by the authors with the analysis of control samples. An additional, important, concern is that while ^{13}C -labelling has the effect of shifting the amide I peak of unlabeled β -sheets from 1632 cm^{-1} to 1587 cm^{-1} , i.e. outside the amide I range for the other conformers (alpha helix etc.) the shifted peak for the isotopically labelled beta sheet (about 1587 cm^{-1}) is within the amide II frequency range of unlabeled peptides. Therefore, personally I would not rely on nor use such analysis for the study of peptides' conformations and, in my opinion, it will not be generally adopted by scientists in this field. Instead, I would trust the well vetted analysis provided by amide I (and possibly amide II) peak position deconvolution to identify the secondary structure of peptide and proteins, which is based on full spectra rather than 2 single frequencies. This approach is widely used in the analysis FTIR and AFM-IR spectra (see ref 26 and several others).

Combining full spectral deconvolution and isotopic labelling, would better achieve the manuscript goals but much more reliably so. Unluckily, state-of-the-art nanoFTIR s-SNOM is not great for recording full spectra because of the low spectral resolution (typically 16 cm^{-1}), long spectral acquisition times (typically 1 to 5 min) and generally low signal to noise. In contrast, AFM-IR spectra, has much higher spectral resolution (typically 1 cm^{-1}), shorter acquisition time (10 to 30 s) and appears better suited for nanoscale conformational analysis of peptides and proteins.

Therefore, in my opinion, the manuscript doesn't provide significant advances to justify the publication in Communications Chemistry and may be better suitable in a very specialized journal (for example Analyst).

Few additional minor comments follow:

- On page 4 "topological" should be "topographic"
- Please specify in the SI the laser power and spectral range used to obtain the nanoFTIR spectra in fig 1.
- The nanoFTIR spectroscopy experimental is not particularly clear. Please double check it. Why if the integration time is set to 10 ms and 5 to 15 interferograms were recorded the total a spectrum acquisition time is 1.5 to 5 min?

27/03/23

Revised version of manuscript COMMSCHEM-22-0526-T

We would like to thank our reviewers for their careful reading of our manuscript and for the challenging comments, which we find improved our manuscript. We have now addressed all comments as detailed below. In the marked manuscript file, text passages that were inserted, edited, or moved are highlighted in yellow. The file *data.zip* was compiled for reviewer 1.

Yours sincerely

Professor Andreas Barth

Department of Biochemistry and Biophysics

Stockholm University
Department of Biochemistry
and Biophysics
Svante Arrhenius väg 16C
SE-106 91 Stockholm, Sweden

Visiting address:
Svante Arrhenius väg 16C
Room K311
<https://www.su.se/english/profiles/abart>

Phone: +46-8-162452
Telefax: +46-8-155597
E-mail: barth@dbb.su.se

Reviewer #1 (Remarks to the Author):

Manuscript untitled: « ¹³C-isotope-editing of nanoscale infrared images reveals the action of an inhibitory peptide against amyloid-β aggregation »

The article presents a new methodology to differentiate 2 peptides with similar secondary structures. For that purpose, the authors have used an already well-known near-field technique (s-snom) and ¹³C labelling peptides (widely used in classical IR spectroscopy). It's an interesting approach, challenging to implement. Unfortunately, the work is not completed in its current form: the methodology is not strong enough. I have a lot of questions and comments but I will only send the most important ones.

1. The introduction should be reinforced, directing it more sincerely towards methodology, especially if this article is to be published in a journal like comm.chem. From my point of view, if the authors insist so much (especially in the title) on the application in biology, it would seem to me more judicious to publish this article in a journal related to biophysics... maybe? But again, not as it is.

We agree with the reviewer and our intention was indeed to write a methodology oriented text. We agree also that the title did not reflect this and have changed it to "¹³C, ¹⁵N-labeling distinguishes different polypeptides in nanoscale images of the infrared absorption. Test case: interaction of the amyloid-β peptide with an inhibitory peptide". We would argue that the introduction section had already been methodology-oriented but admittedly with a focus on polypeptides and thus with applications in biochemistry in mind. Our arguments for choosing *Communications Chemistry* were that the method belongs to spectroscopy, which is a topic in Physical Chemistry, and the studied material is from Biochemistry. In response to the comment, we have now lifted the approach of isotope-editing of the nanoscale images up and discuss it in more detail at the beginning of the introduction with a broader view on the application range. This section follows the introduction of nanoscale infrared spectroscopy and reads now:

"The technique is particularly useful to study mixtures, which are of interest in diverse fields of chemistry: e.g. in material science, nanotechnology, biochemistry, and medical chemistry. Such mixtures may be composite materials, self-assembled molecular structures, multi-component colloids, or blends of biomolecules – either in their natural context or mixed in the test tube – which are of interest for their biological function or for engineering new compounds with designed properties.

Nanoscale infrared spectroscopy can identify the location of each mixture component and thus reveal its morphology, as long as the chemical structures of the mixture components are sufficiently distinct to generate infrared absorption bands in different regions of the spectrum. However, chemically similar molecules generate similar infrared spectra and are therefore difficult to distinguish. In such cases, it is useful to label one of the mixture components with a stable isotope. The different mass of the labeled component shifts the spectral position of its absorption bands and makes its spectrum distinguishable from that of the unlabeled compound(s). This approach has been used in infrared nanospectroscopy to distinguish synthetic polymers by deuteration,^{2,3} which generates a large isotope shift and moves the absorption band of the deuterated groups into a spectral region that is largely clear from the absorption of other groups. "

We continue then with the particular example of polypeptides and proteins.

We further enhanced the methodological aspect by discussing in *Results and Discussion* the amide II band in considerable detail and the noise in the spectra (see comment 5), as well as by extending *Conclusion*, which

now mentions particular challenges in using ^{13}C , ^{15}N -labeling for peptides, highlights the feasibility of polypeptide labeling, and discusses advantages of the approach. These sections are pasted below.

Results and Discussion:

"The amide II band is found at 1553 cm^{-1} (average of 5 spectra, standard error 2 cm^{-1}) which is 10-20 cm^{-1} higher than expected for antiparallel β -sheet structures^{53,54} but in line with previous A β spectra.⁵⁵⁻⁵⁸ Its position testifies to the formation of parallel β -sheets in the transition from A β oligomers to fibrils.⁵⁶

The amide II band is relatively strong compared to the amide I band, which is likely due to a polarization effect. The near field under the AFM tip is polarized perpendicular to the sample surface. Thus transitions with a transition dipole moment (TDM) that is perpendicular to the surface will absorb strongly, but those with a parallel TDM only weakly. For β -sheets, the main transition is polarized perpendicular to the strand direction.⁵⁹⁻⁶¹ Since the strand direction is perpendicular to the fibril axis for amyloid fibrils, the TDM is always parallel to the surface and absorbs weakly. In contrast, the TDM of the amide II vibration is largely parallel to the strand direction,^{59,60} which gives rise to considerable absorption whenever the strand direction is not parallel to the sample surface. This is the case because the amyloid fibrils are twisted.⁶² Therefore we expect a stronger relative amide II contribution to the nano-FTIR spectra than to solution FTIR spectra of bulk samples. The opposite effect has been observed for the purple membrane,⁴ which has the α -helical protein bacteriorhodopsin as a main constituent. Here, the amide I band is much stronger than the amide II band because the amide I TDM is perpendicular to the sample surface, whereas the amide II TDM is parallel to the surface. "

"Our spectra of labeled A β 40 exhibit a reduced amide II absorption relative to the amide I band (Fig. 1 and Fig. S2). This can be tentatively explained by the effects of isotope-labeling, which is known not only to affect the vibrational frequency of a normal mode, but also to alter its internal coordinate contributions – with possible consequences for the direction and magnitude of its TDM. Density functional theory calculations for an amide model compound⁷⁰ indicate only small changes of the TDM orientation upon either ^{13}C - or ^{15}N -labeling. The magnitude of the amide I transition dipole moment is slightly reduced by ^{13}C -labeling (4%) but not by ^{15}N -labeling. However, both isotopes reduce the magnitude of the amide II TDM: ^{13}C -labeling by 9% and ^{15}N -labeling by 13%. Assuming additivity of the isotope effects on the amide II TDM magnitude and considering that the integrated absorbance is proportional to the squared TDM magnitude, we estimate a reduction of the ^{13}C , ^{15}N -amide II band by 38% relative to the unlabeled amide II band and a corresponding reduction of the amide I band by 9%. Thus, the amide II band is expected to be considerably weaker relative to the amide I band for the labeled peptide than for the unlabeled peptide, which is in line with our experimental spectra."

Conclusions:

"Due to the small isotope shift of $\sim 40\text{ cm}^{-1}$ and the absence of a clear spectral window for the amide I band of ^{13}C - or ^{15}N -labeled polypeptides, the approach is challenging to implement."

"In future applications, full-range spectra at locations of interest identified by the isotope-edited images will be a valuable complement to the morphology information of the individual interaction partners. While it even would be desirable to have full spectral information for the entire area of nanoscale images (hyperspectral imaging), monitoring at a few selected wavenumbers dramatically speeds up the recording of the images and thus minimizes the risk of tip contamination or tip damage during the experiment. It will enable researchers to scan a larger sample area than with hyperspectral imaging leading to a more representative sampling of the sample properties.

Such imaging of spectral properties at particular wavenumbers or wavelengths has been proven valuable in infrared and Raman microscopy to map particular functional groups and in fluorescence microscopy to map one or several specific molecules. Similar to fluorescence microscopy, the present approach can identify different polypeptides, but has two advantages: (i) it beats the diffraction limit and enables mapping with 20 nm spatial resolution, and (ii) it avoids the need to attach a bulky label for monitoring, which might affect the chemical properties and modify the interactions to be studied.

For very many applications, isotope labeling does not present an obstacle because it can be achieved either by expression of recombinant, labeled proteins or by chemical synthesis of labeled peptides. Studying the interaction of one particular polypeptide – e.g. A β – with several other partners can be done in an economic way as it requires only to label the common polypeptide of interest (A β), while each of the other interaction partners may remain unlabeled in binary mixtures. The approach can be extended to

more than two interacting polypeptides by imaging one labeled interaction partner at a time in a series of multi-component mixtures. "

The authors need to argue more:

2. Why did they choose this NCAM1-Prp hybrid peptide for interactions with A β 40-42? Even if it's clearly described in their previous paper, we need a short reminder. What puzzles me more is the following: some research groups have shown that the species bound to PrPC are predominantly small oligomers (not only fibrils). Would not it be the same for the hybrid peptide: as the authors have the technology to study molecular interaction at the fibrillar level why not explore this aspect?

The anti-amyloid action of the NCAM1-PrP peptide was discovered by my colleague Astrid Gräslund by a mixture of insight and serendipity. She first realized that the N-terminal prion protein sequences (including signal peptides) have similarities with cell-penetrating peptides (CPPs), showed that such sequences are indeed CPPs (Lundberg et al. 2002, BBRC 299, 85-90, Magzoub et al. 2006, Biophys. Res. Comm. 348, 379-385), and then found that these CPPs hindered prion propagation (Löfgren et al. 2008, FASEB J. 22, 2177-2184; Löfgren Söderberg et al. 2014, Arch. Biochem. Biophys. 564, 254-261). This triggered the interest to make use of the anti-amyloid properties of the peptide. For this it was tested whether the signal peptide sequence of the prion protein can be exchanged by the shorter sequence from the NCAM1 protein that also signals for secretion. Indeed, the same anti-prion activity was observed (Löfgren Söderberg et al. 2014, Arch. Biochem. Biophys. 564, 254-261) and the next step was to test whether the peptide is active also against other amyloid proteins, like the amyloid- β (A β) peptide. It could be shown that the NCAM1-PrP peptide inhibited A β amyloid formation and relieved A β induced neurotoxicity (Henning-Knechtel et al. 2020, Cell Reports Physical Sci. 1, 100014). This is now better described in the introduction:

"NCAM1-PrP was designed by some of us based on the discovery that the N-terminal portion of the nascent prion protein sequence has anti-prion activity.³⁹ In NCAM1-PrP, the prion signal sequence that targets the protein for the secretory pathway was replaced by the equivalent, but shorter, signal sequence from the neural cell adhesion molecule-1 (NCAM1, residues 1-19: MLRTKDLIWLFFLGTAVS). The C-terminal part of NCAM1-PrP consists of the positively charged hexapeptide KKRPKP, which corresponds to residues 23-28 of the nascent mouse prion protein sequence. NCAM1-PrP aggregates on its own at concentrations of 20 μ M and more and forms β -sheet structure in the process. This results in a prominent β -sheet band in the infrared spectrum with a spectral position of 1622 cm^{-1} in D₂O.⁴⁰ (Note that Fig. 3 in the cited reference is mislabeled. The red spectrum is that of NCAM1-PrP). NCAM1-PrP has not only anti-prion activity but inhibits also the aggregation of A β in vitro⁴⁰⁻⁴³ by targeting⁴⁰ the dominant mechanism for A β fibril formation: fibril catalyzed formation of new fibrils (secondary nucleation).^{44,45} It also counteracts the lethal damage inflicted by the A β peptide on neuroblastoma cells.⁴² The cationic hexapeptide⁴² and the signal sequence^{39,41} are both important for the anti-amyloid activity and the latter seems to target a relevant cellular location in the neuronal cells.^{41,42} These properties of NCAM1-PrP make it a promising starting point for developing a treatment against AD."

We note that the NCAM1-PrP peptide contains only 6 residues from the prion protein. Therefore it cannot be expected to have the same interactions as PrP^c. But definitely, the interaction with A β oligomers is an interesting topic and on our to-do list. For the present work however, we focused on fibrils because of their stronger signals (see below).

3. Furthermore, PrP 23–27 region is located within the part of the prion protein that is natively unstructured. If we look at their previous paper (ref 39), compared to A β 42, the IR absorption spectrum of this peptide does not exhibit a high absorption at 1629 cm^{-1} but a broad band centred around 1645-1640 cm^{-1} . So why the authors only acquired the 1629 cm^{-1} IR map?

How do the authors justify the lack of spectral data about this NCAM1-Prp hybrid (I mean in the present paper)? This point is crucial for the consistency of the paper. It's a serious methodological issue.

We would like to thank the reviewer for this comment, because it has revealed a mistake in the Król et al. article (old ref 39): The top spectra in both panels of Fig. 3 are mislabeled, whereas the description in the text is correct. The blue spectra are those of A β 40 and the red spectra those of NCAM1-PrP. Thus, the NCAM1-PrP peptide has a prominent band at 1622 cm⁻¹ in the β -sheet region, whereas A β 40 is partly unstructured. We will correct this mistake and have now discussed these spectra in *Introduction* (see quotation related to comment 2). The NCAM1 portion of NCAM1-PrP aggregates and forms fibers on its own (Henning-Knechtel et al. 2020) and its properties may thus override the intrinsic properties of the PrP hexapeptide sequence.

4. Then, spectral measurements were limited to bundles of fibres whereas we are waiting for measurements on individual fibres. Is there a reason? It's almost the same problem for imaging. The interpretation of the signal is more or less clear for bundles but not for singles fibres. The images should clarify this point but as there are a lot of bundles it's impossible to conclude even when looking carefully at supplementary figures.

Our impression is that we are on the limit of what is possible to measure. The quality of the spectra depends dramatically on the amount of material under the tip and on the spatial extension of the sample. Thus, our 2D purple membrane sample produced much better spectra (now taken out from the SI) than 1D (linear) samples like isolated fibers, which are in turn easier to study than point samples like small oligomers. A consequence of this is that fiber bundles have a better signal to noise ratio than individual fibers and for that reason we focused on fiber bundles for the spectra. Still, the quality of the spectra cannot be compared with ordinary FTIR spectra of bulk samples (this is further discussed in the answer to the next comment) but this is not surprising given the dramatically smaller amount of material that is measured.

Our instrument is essentially a combination of two separate instruments: one for spectrum recording and one for imaging. Light source, optical path and detector are different for the two operation modes, the only common elements are the AFM part and the mirror that focuses the laser beam on the tip. In spectrum recording we use a broad band laser, whereas in imaging we use a monochromatic laser. This implies more light intensity per wavenumber interval in the imaging mode and thus better signals as was already mentioned in the original text.

We judge that there are many individual fiber segments in our A β 40 images (Fig. 2) and most of them are very thin. We claim that we can see the IR absorption of very thin fibers (< 2 nm high) in the images and even the expected differences between the two wavenumbers for labeled and unlabeled peptides. This is also evident in the line profiles in Figs. S4 and S5, where fibrils with heights smaller than 2 nm can be detected unequivocally by their optical phase.

5. From the spectra it seems that there are a lot of variabilities. How do the authors explain this observation? Why do we observe so many oscillations on spectra 1 (reference)? Why is the amide II band more intense than the amide I band for the unlabeled A β but not for the labelled A β ? Still, we observe a lot of oscillations on the IR spectra, why?

We were also initially puzzled by the oscillations and discussed them with the Neaspec experts. We think that the oscillations are caused by the noise. We record the spectra at the low resolution of 16 cm^{-1} in order to increase the signal to noise ratio. However, this implies only a few data points in the amide I region where the noise then can manifest as oscillation. We discuss the oscillations now in the text:

"We note that all spectra seem to contain oscillations, which is in contrast to the expected smooth appearance of the protein and the control spectra. We tentatively ascribe the "oscillations" to the noise. Noise leads to a deviation of the measured data points from the correct spectrum. Since the wavenumber interval between two data points is 16 cm^{-1} at the spectral resolution used (ignoring the data points introduced by zero-filling), the noise will generate "ups and downs" in the measured spectrum with a spacing of 16 cm^{-1} . This gives the impression of an oscillation with a periodicity of 32 cm^{-1} , corresponding to twice the resolution. This is in fact the main "oscillation" period observed in the data."

From our control spectra of the purple membrane spectra we conclude that our spectra have the same quality as previously published spectra and therefore that the oscillations are not due to some experimental mistake.

Regarding the amide II band we discuss now the strength of this band relative to the amide I band for the unlabeled sample and give an explanation for its relative weakness for the labeled peptide. To facilitate this discussion we show the averaged spectra from Fig. 1 in the new Fig. S2.

The new text regarding the unlabeled peptide spectrum in the main manuscript reads:

"The amide II band is relatively strong compared to the amide I band, which is likely due to a polarization effect. The near field under the AFM tip is polarized perpendicular to the sample surface. Thus transitions with a transition dipole moment (TDM) that is perpendicular to the surface will absorb strongly, but those with a parallel TDM only weakly. For β -sheets, the main transition is polarized perpendicular to the strand direction.⁵⁶⁻⁵⁸ Since the strand direction is perpendicular to the fibril axis for amyloid fibrils, the TDM is always parallel to the surface and absorbs weakly. In contrast, the TDM of the amide II vibration is largely parallel to the strand direction,^{56,57} which gives rise to considerable absorption whenever the strand direction is not parallel to the sample surface. This is the case because the amyloid fibrils are twisted.⁵⁹ Therefore we expect a stronger relative amide II contribution to the nano-FTIR spectra than to solution FTIR spectra of bulk samples. The opposite effect has been observed for the purple membrane,⁴ which has the α -helical protein bacteriorhodopsin as a main constituent. Here, the amide I band is much stronger than the amide II band because the amide I TDM is perpendicular to the sample surface, whereas the amide II TDM is parallel to the surface."

The new text regarding the labeled peptide spectrum in the main manuscript reads:

"Our spectra of labeled A β 40 exhibit a reduced amide II absorption relative to the amide I band (Fig. 1 and Fig. S2). This can be tentatively explained by the effects of isotope-labeling, which is known to not only affect the vibrational frequency of a normal mode, but also to alter its internal coordinate contributions, with possible consequences for the direction and magnitude of its TDM. Density functional theory calculations for an amide model compound⁶⁷ indicate only small changes of the TDM orientation upon either ^{13}C - or ^{15}N -labeling. The magnitude of the amide I transition dipole moment is slightly reduced by ^{13}C -labeling (4%) but not by ^{15}N -labeling. However, both isotopes reduce the magnitude of the amide II TDM: ^{13}C -labeling by 9% and ^{15}N -labeling by 13%. Assuming additivity of the isotope effects on the amide II TDM magnitude and considering that the integrated absorbance is proportional to the squared TDM magnitude, we estimate a reduction of the ^{13}C , ^{15}N -amide II band by 38% relative to the unlabeled amide II band and a corresponding reduction of the amide I band by 9%. Thus, the amide II band is expected to be considerably weaker relative to the amide I band for the labeled peptide than for the unlabeled peptide, which is in line with our experimental spectra."

6. What kind of data processing was done on the spectra? It would be useful to have access to the raw spectra.

The spectra were only corrected for the phase tilt and a phase offset as described already in the submitted version of the SI ("Phase tilt and phase offset were corrected and the nanoFTIR absorption^{1,2} calculated with

the neaPLOT software... "). We will make the raw data available upon acceptance of the manuscript but upload in the revision those for the spectra so that they can be inspected (file *data.zip*). The raw data have the extension *txt* and contain a column with the wavenumbers and columns with the amplitude and phase data at different harmonics of the tip oscillation frequency. We used amplitude O2A and phase O2P at the second harmonic to calculate the nano-FTIR absorption, which is the imaginary part of $O2A \cdot \exp(i \cdot O2P)$. This was calculated with the neaPLOT program (neaSPEC, Germany), which is no longer freely available. However, the phase spectrum O2P is similar to the nano-FTIR absorption spectrum for small phase values as in our case and can be extracted from the raw data. We include also the processed data (extension *csv*), which are easier to inspect. The file name indicates the spectrum number in Fig. 1. *imag* indicates the nano-FTIR absorption spectrum and *phase* the phase spectrum. We also show here the unprocessed nano-FTIR (top) and phase (bottom) data for the unlabeled A β 40.

Fig. 1 Unprocessed nano-FTIR (top) and phase (bottom) data for the unlabeled A β 40. The color code is the same as in Fig. 1 of the main text.

7. Figure 2, 3, 4: Why the authors do not show the optical amplitude at 1587 cm⁻¹?

The optical amplitude is useful to visualize the sample but less so for the spectral interpretation. The optical amplitude data at 1587 cm⁻¹ look essentially like those at 1629 cm⁻¹. This is now mentioned in the text:

"Column C shows images of the near-field amplitude of the scattered radiation, which is related to the reflectivity of the sample. It is an additional way to visualize the sample, but not used to interrogate its spectral properties. Therefore, we show only the optical amplitude at one of the two wavenumbers and we arbitrarily selected that at 1629 cm⁻¹."

I paste below screenshots for all images where the unprocessed optical amplitude at 1629 cm⁻¹ is on the left hand side and that at 1587 cm⁻¹ on the right hand side.

Fig. 2. Unprocessed optical amplitude at 1629 cm⁻¹ (left) and at 1587 cm⁻¹ (right).

8. Figure 4 top panel: the hybrid peptide seems to exhibit a low signal at 1629 cm^{-1} . Why the signal is higher when it's mixed with the labelled A β 40?

The NCAM1-PrP fibers are much thinner than the fibers in the mixed sample (the color scale in the topography images extends to a height of 6 nm for the NCAM1-PrP peptide and to 25 nm for A β 40 in Fig. 4). Many of the fibers in the pure NCAM1-PrP sample are 3 nm high or less (see Table S1). Such fibers exist also in the mixed sample (Table S1) but the optical impression of its phase image is dominated by the strongly absorbing, thick fibers. Most important in the context of our work is that the optical phase of the NCAM1-PrP fibers at 1629 cm^{-1} is larger than at 1587 cm^{-1} , as expected for an unlabeled peptide.

9. In the conclusion, the authors claim that “These observations indicate that NCAM1-PrP is able to coat or to dissolve existing A β 40 fibrils”: I don't understand how the authors arrive at this conclusion just by looking at figure 4 and S4.

We have now considerably expanded our text regarding the A β :NCAM1-PrP sample and hope that our arguments have become clearer. The new text reads:

"The results of mixing fibrillar NCAM1-PrP and labeled, fibrillar A β 40 are shown in the bottom row of Fig. 4 and the lower half of Fig. S5. The latter figure shows also an additional sample area compared to Fig. 4. If the two peptides were not interacting, we would expect aggregates with either a stronger signal at 1629 cm^{-1} (for NCAM1-PrP) or at 1587 cm^{-1} (for A β 40). Since the pure A β 40 samples were rich in fibrils, we would expect also pure A β 40 fibrils in the mixed sample, *i.e.* fibrils with a stronger signal at 1587 cm^{-1} . However, neither the difference images (Fig. 4 and Fig. S5) nor the line profiles (Fig. S5) indicate pure A β 40 fibrils: there are no red fibrils in the difference images of the A β 40:NCAM1-PrP sample that would indicate stronger absorption at 1587 cm^{-1} than at 1629 cm^{-1} and none of the fibrils examined by the line profiles (Fig. S5) exhibits a stronger absorption at 1587 cm^{-1} (red curves) than at 1629 cm^{-1} (blue curves) as would be expected for labeled A β 40. This is in contrast to the pure, labeled A β 40 sample, where all examined fibrils display this behavior (Fig. S4). It is also in contrast to the mixture of labeled and unlabeled A β 40 and the A β 40:A β 42 mixture (Figs. 2, 3, and S5), where fibrils with stronger signal at 1587 cm^{-1} can be observed (see Fig. S5, *e.g.* lines 3, 4, and 5 for the mixture of labeled and unlabeled A β 40, and lines 4, 6, 7, and 10 for the mixture of labeled A β 40 and unlabeled A β 42). This demonstrates that labeled fibrils can be detected in mixtures of labeled and unlabeled peptides. For the A β 40:NCAM1-PrP sample we conclude that there is no indication of pure A β 40 fibrils in the presence of NCAM1-PrP. Regions that can be associated with A β 40 (red areas) in the difference images stem instead from amorphous aggregates (Fig. 4 and Fig. S5).

The fibrils of the A β 40:NCAM1-PrP sample show either equal absorption at 1587 and 1629 cm^{-1} in the difference images (gray), characteristic for a mixed composition, or stronger absorption at 1629 cm^{-1} (blue), characteristic for the unlabeled peptide NCAM1-PrP. Similarly, in the line profiles (Fig. S5), the examined fibrils have either similar absorption at 1587 cm^{-1} (red curves) and at 1629 cm^{-1} (blue curves) or stronger absorption at 1629 cm^{-1} (blue curves). The fibrils in our images are therefore either composed of both peptides or predominately of NCAM1-PrP.

Since we did not observe pure A β 40 fibrils, which were present before mixing, we conclude that mixing with NCAM1-PrP either transformed them to amorphous aggregates or that they became a constituent of fibril co-assemblies. This is not due to the mixing process itself, as labeled A β 40 fibrils were observed in the mixture of labeled and unlabeled A β 40 and in the mixture of labeled A β 40 and unlabeled A β 42. Thus, we propose that the observed effects are due to the action of NCAM1-PrP, which associated with or dissolved existing A β 40 fibrils. It remains to be revealed, which of the NCAM1-PrP species – monomers, oligomers, or fibrils – is responsible for the effect.

Our interpretation is in line with previous findings. NCAM1-PrP was found to inhibit secondary nucleation of A β in a similar way as BRICHOS,⁴⁰ which is known to coat fibrils and thus prevents the surface-catalyzed formation of new fibrils.⁷⁸ The analogous action of NCAM1-PrP thus suggests that it can bind to A β fibrils and this will be facilitated by the opposite charges of the two peptides. Binding of

NCAM1-PrP to A β fibrils is supported by our observation of fibrils with similar absorption at 1587 cm⁻¹ and at 1629 cm⁻¹ indicating that they consist of both A β and NCAM1-PrP. Such heterogeneous fibrils are conceivable, since fibrils formed from two different peptides have been discovered previously.⁷⁹⁻⁸¹

We observed amorphous A β 40 aggregates but no pure A β 40 fibrils after addition of NCAM1-PrP although the A β sample contained fibrils. Also in the interaction with monomeric A β , NCAM1-PrP prevents the formation of amyloid fibrils and produces amorphous aggregates.^{40,42} A second finding in this context is that NCAM1-PrP promotes the aggregation of the protein S100A9 by destabilizing the native structure and creating a misfolded state.⁷⁷ It has therefore been suggested that NCAM1-PrP unfolds or dissolves structured proteins by interaction with amyloidogenic sequences,⁴⁰ which is in line with our proposal that NCAM1-PrP is able to destabilize A β fibrils. "

10. In the SI: the part related to purple membrane is completely irrelevant. The authors should remove it.

The purple membrane section was originally included to demonstrate the performance of our instrument with the used settings. We agree that it is otherwise not related to the rest of the manuscript and have taken it out.

Reviewer #2 (Remarks to the Author):

The manuscript by Paul et al. entitled "13C-isotope-editing of nanoscale infrared images reveals the action of an inhibitory peptide against amyloid- β aggregation" described for the first time that 13C labelled proteins or peptides (here on AB fibrils) can be observed and distinguished from the same non-labelled proteins by nano-IR. This is of importance when comparing the same protein or proteins with similar secondary structure. Furthermore, they showed that such labelled spectral features can be observed on sample as thin as 2 nm. Finally, they use their method to study the interaction of the inhibitory NCAM1-Prp on AB aggregation. This is an important addition to both the field of nano-IR and amyloid study.

Thank you for this judgment.

On page 4, line 10: the authors gave the minimal (2nm) and maximal (12 nm) thickness of the Ab40 fibrils. It would be interesting to know what the average size of the fibrils is. Does it correspond to regular amyloid fibrils (7-10 nm)?

We have included a more comprehensive analysis of our fibrils and compare our measured heights with literature AFM data. The new text reads:

"An analysis of the heights of 43 fibril segments showed that 16% were thinner than 2.0 nm (smallest height: 0.5 nm), 26% between 2.0 and 2.8 nm, and ~20% each had height ranges of 3.4-4.2 nm, 4.4-5.6 nm, and 6.0-10.2 nm. The height range of the fibrils in our study agrees with that of previous studies, which have identified several A β 40 fibril types with heights between 0.65 to 12 nm.⁷¹⁻⁷⁴ Fibrils with heights around 2 nm were found to be composed of 2 filaments, those with heights of ~4 nm of 4-6

filaments. More than 6 filaments increase the height only little.⁷³ When compared with electron microscopy images, the AFM-derived widths are larger even in high resolution AFM measurements that use tips with ~ 1 nm diameter, while the heights are considerably smaller. The latter may be explained by the arrangement of filament pairs on the substrate,⁷³ the effect of different electrostatic interactions of the tip with the sample and the substrate⁷⁵ or the compression of soft biomolecules caused by the interaction with the AFM tip.^{71,76,77}

On page 4, lines 21-22: "There is non-fibrillar material on the top of these images, which absorbs stronger at 1587 cm^{-1} than at 1629 cm^{-1} and which we attribute to an impurity because of its different shape." In almost all the images, there is high absorbance at 1587 cm^{-1} in globular shape entities. How to distinguish these from the potential impurity mentioned before? Or the globulomers (potentially oligomers) contains more labelled peptides than fibrils. This is never mentioned or explained.

Our text refers to the top row of images in Fig. 2, which shows results for the unlabeled A β peptide. For this peptide we do not expect a stronger absorption at 1587 cm^{-1} than at 1629 cm^{-1} for any of the aggregates. We were puzzled by this until we realized that the particle in question has some unusual features, as better described in our revised text:

"We attribute this material to an impurity because it has a well defined shape and is rather isolated from other material. In particular, it is well separated from fibrillar structures. In contrast, amorphous material of similar dimensions is well-integrated into a network of fibrils in all other images (see Figs. 2-4)."

The figure below is a part of Fig. 2 of the main text, which is pasted here to illustrate the argument above.

Fig. 3. Copy of part of Fig. 2 of the main text. Left: topography, middle: mechanical phase, right: optical amplitude at 1629 cm^{-1} .

On page 5, lines 4-5: "In many cases, the image showed a large predominance of either the labeled or the unlabeled peptide". Is there an explanation for this, except for the absence of vortexing?

A tentative explanation is now included in the text:

"We speculate that this is due to the repulsion between extensive networks of negatively charged fibrils, which impedes a homogeneous mixing of labeled and unlabeled peptides. Because of the small area of our images, most of them contained material from either a labeled or an unlabeled network of fibrils. It might be argued against this explanation that the formation of fibrils from A β monomers is thermodynamically favorable in spite of the electrostatic repulsion between the A β peptides. However, this might either not apply to the association of already formed fibrils or the time between mixing and sample preparation was too short to allow fibril association to proceed to a significant extent."

On page 5, lines 13-15: the authors indicated the color code to understand the difference spectral images (last column of fig 2, 3 and 4 and supplementary figures), it would be clearer to indicate these in the legend of fig 2, the first time the authors showed these images.

We agree to the comment and have inserted the following text at the end of the legend of Fig. 2 and we have referred to this figure legend in the other figure legends.

"Positive values are coded in red and indicate more absorption at 1587 cm⁻¹ than at 1629 cm⁻¹, which is characteristic of the labeled peptide. Negative values are shown in blue and reveal more absorption at 1629 cm⁻¹, which is characteristic of the unlabeled peptide. Phase differences close to zero are shown in white. The scale of the phase difference image is symmetrical around zero, *i.e.* the absolute values of the maximum and minimum scale values are the same. The overlaid semi-transparent height image indicates the height in a gray scale. Gray regions without a blue or red color tone indicate peptide aggregates where the phase difference is close to zero."

On page 5, line 25: "Therefore, it is advisable to inspect additionally a plot of the phase along lines that run across structural features of interest as shown in Fig. S3. » When looking at fig. S3 and S4, spectral plots with Ab42 (and NCAM1-PrP in S3) displayed narrower bands than with Ab40, is there an explanation for this?"

There may be two reasons for the impression that the A β 42 and NCAM1-PrP features are narrower than the those of A β 40 in Fig. S3 (now Fig. S4). (i) The length scale of the line profile plots is larger for the former peptides because longer lines are monitored. This gives an impression of narrower features. (ii) The NCAM1-PrP sample and the A β 42 protofibril sample contained smaller/thinner aggregates throughout, whereas the fibrillar A β 40 samples contain a mixture of large and small features, where the large features dominate the visual impression. These differences between the different samples are due to the different properties of the peptides and the different preparation protocols.

In Fig. 1, corresponding to the explanation on page 6 lines 14-18, it is not clear why the spectral features of the labelled peptide are not present at all in the bottom row. Therefore, how was it possible to calculate the 1587 minus 1629 cm-1 images if in spectra from a mixture of labelled and non-labelled peptides the band at 1587 cm-1 is absent?"

There is actually one spectrum that shows features expected for the dominance of labeled peptide. It is spectrum 2, taken in a red area of the difference image. The amide I absorption maximum is clearly below 1600 cm⁻¹ and coincides with the absorption maximum for labeled peptides. In the other spectra we were unlucky in picking the locations for recording the spectra. It is difficult identify locations with predominantly labeled peptide in the phase images without careful processing of the data. This and generation of the difference images takes time and can therefore not be done during a measurement. Therefore, choice of locations for spectrum recording was suggested by the topography of the sample. The result of this

unfortunate picking of locations was that we recorded one spectrum with predominantly unlabeled peptide but all other locations had a mixture of labeled and unlabeled peptides. The spectrum at location 2 has most contribution from the labeled peptide and therefore its maximum is close to 1590 cm^{-1} .

The spectral quality of the images is better than that of the spectra because a broad band laser is used in the latter approach and a monochromatic laser in the former. This results in a higher light intensity per wavenumber interval for imaging and a better signal to noise ratio.

Regarding our choice of the wavenumbers for imaging, we argue that it can be motivated by the spectra of the pure samples, shown in Fig. 1 and now also in Fig. S2 for the averaged spectra. These spectra show clearly the expected features of labeled and unlabeled peptides and their band positions coincide with those found in the literature. This is now discussed in considerable detail at the beginning of *Results and Discussion*.

On page 7, the explanation of the role of NCAM1-PrP is difficult to follow. First, the authors stated that none of the mixed fibrils exhibits a stronger absorption at 1587 cm^{-1} . Thus, the formed fibrils should be composed of NCAM1-Prp and not labelled Ab40. But later the authors wrote "the mixed composition of the fibrils in our Ab:NCAM1-Prp images ...". For us there is no mixed composition of fibrils, once more all the labelled peptides seem to be in globular shape features.

We have now considerably expanded our interpretation and hope that it is clearer now. We also added references to studies that have discovered fibers with mixed composition. The new text reads:

"The results of mixing fibrillar NCAM1-PrP and labeled, fibrillar A β 40 are shown in the bottom row of Fig. 4 and the lower half of Fig. S5. The latter figure shows also an additional sample area compared to Fig. 4. If the two peptides were not interacting, we would expect aggregates with either a stronger signal at 1629 cm^{-1} (for NCAM1-PrP) or at 1587 cm^{-1} (for A β 40). Since the pure A β 40 samples were rich in fibrils, we would expect also pure A β 40 fibrils in the mixed sample, *i.e.* fibrils with a stronger signal at 1587 cm^{-1} . However, neither the difference images (Fig. 4 and Fig. S5) nor the line profiles (Fig. S5) indicate pure A β 40 fibrils: there are no red fibrils in the difference images of the A β 40:NCAM1-PrP sample that would indicate stronger absorption at 1587 cm^{-1} than at 1629 cm^{-1} and none of the fibrils examined by the line profiles (Fig. S5) exhibits a stronger absorption at 1587 cm^{-1} (red curves) than at 1629 cm^{-1} (blue curves) as would be expected for labeled A β 40. This is in contrast to the pure, labeled A β 40 sample, where all examined fibrils display this behavior (Fig. S4). It is also in contrast to the mixture of labeled and unlabeled A β 40 and the A β 40:A β 42 mixture (Figs. 2, 3, and S5), where fibrils with stronger signal at 1587 cm^{-1} can be observed (see Fig. S5, *e.g.* lines 3, 4, and 5 for the mixture of labeled and unlabeled A β 40, and lines 4, 6, 7, and 10 for the mixture of labeled A β 40 and unlabeled A β 42). This demonstrates that labeled fibrils can be detected in mixtures of labeled and unlabeled peptides. For the A β 40:NCAM1-PrP sample we conclude that there is no indication of pure A β 40 fibrils in the presence of NCAM1-PrP. Regions that can be associated with A β 40 (red areas) in the difference images stem instead from amorphous aggregates (Fig. 4 and Fig. S5).

The fibrils of the A β 40:NCAM1-PrP sample show either equal absorption at 1587 and 1629 cm^{-1} in the difference images (gray), characteristic for a mixed composition, or stronger absorption at 1629 cm^{-1} (blue), characteristic for the unlabeled peptide NCAM1-PrP. Similarly, in the line profiles (Fig. S5), the examined fibrils have either similar absorption at 1587 cm^{-1} (red curves) and at 1629 cm^{-1} (blue curves) or stronger absorption at 1629 cm^{-1} (blue curves). The fibrils in our images are therefore either composed of both peptides or predominately of NCAM1-PrP.

Since we did not observe pure A β 40 fibrils, which were present before mixing, we conclude that mixing with NCAM1-PrP either transformed them to amorphous aggregates or that they became a constituent of fibril co-assemblies. This is not due to the mixing process itself, as labeled A β 40 fibrils were observed in the mixture of labeled and unlabeled A β 40 and in the mixture of labeled A β 40 and unlabeled A β 42. Thus, we propose that the observed effects are due to the action of NCAM1-PrP, which associated with or

dissolved existing A β 40 fibrils. It remains to be revealed, which of the NCAM1-PrP species – monomers, oligomers, or fibrils – is responsible for the effect.

Our interpretation is in line with previous findings. NCAM1-PrP was found to inhibit secondary nucleation of A β in a similar way as BRICHOS,⁴⁰ which is known to coat fibrils and thus prevents the surface-catalyzed formation of new fibrils.⁸² The analogous action of NCAM1-PrP thus suggests that it can bind to A β fibrils and this will be facilitated by the opposite charges of the two peptides. Binding of NCAM1-PrP to A β fibrils is supported by our observation of fibrils with similar absorption at 1587 cm⁻¹ and at 1629 cm⁻¹ indicating that they consist of both A β and NCAM1-PrP. Such heterogeneous fibrils are conceivable, since fibrils formed from two different peptides have been discovered previously.⁸³⁻⁸⁵

We observed amorphous A β 40 aggregates but no pure A β 40 fibrils after addition of NCAM1-PrP although the A β sample contained fibrils. Also in the interaction with monomeric A β , NCAM1-PrP prevents the formation of amyloid fibrils and produces amorphous aggregates.^{40,42} A second finding in this context is that NCAM1-PrP promotes the aggregation of the protein S100A9 by destabilizing the native structure and creating a misfolded state.⁷⁷ It has therefore been suggested that NCAM1-PrP unfolds or dissolves structured proteins by interaction with amyloidogenic sequences,⁴⁰ which is in line with our proposal that NCAM1-PrP is able to destabilize A β fibrils. "

In the legend of fig 4, point E: it is not "optical phase at 1587 cm-1" but 1629 cm-1

We hope that there is no misunderstanding, but we have checked and find our annotation of column E correct. But there was a mistake in the text for column D that has been corrected. In Figs. 2 to 4, panels D show the optical phase at 1629 cm⁻¹, panels E the optical phase at 1587 cm⁻¹ and panels F the phase differences. The difference images indicate that the panels were not confused: In the top difference image of Fig. 4 (unlabeled NCAM1-PrP) we expect stronger absorption at 1629 cm⁻¹, which is in line with the stronger absorption in panel D than in panel E.

Reviewer #3 (Remarks to the Author):

In "13C-isotope-editing of nanoscale infrared images reveals the action of an inhibitory peptide against amyloid- β aggregation" Paul et al. used infrared (IR) scattering-type near-field optical microscopy (s-SNOM) to map the distribution of natural and 13C-labelled amyloid- β peptides deposited on a substrate. s-SNOM is one of few of AFM-based nanoscale IR spectroscopy measurement paradigms which have been widely used for material characterization in the recent literature. Analysis of the conformational arrangement of such amyloids at the nanoscale is important because amyloids aggregates are linked to Alzheimer's and other diseases and because such aggregates are typically both morphologically and conformationally heterogeneous at the nanoscale. Therefore, nanoscale conformational analysis of such aggregates is being studied extensively.

Although selective isotopic labelling is a widely used strategy in IR spectroscopy to distinguish components within mixtures of otherwise spectrally similar molecules and has been used previously in nanoscale IR applications (see for example ref #33 and Analyst, 2018, 143, 3808–3813). I believe this is the first time it has been employed to study peptides at the nanoscale. I should be noted that isotopic labelling is not universally useful, and it is rarely used in samples

composed by initially unknown species but may be effective in selected applications, often where the average composition of the sample is known a priori.

Thank you for pointing us to the interesting reference which has escaped our literature search. We cite it now in the introduction. We agree to the judgment of the usefulness and have added that the approach will be useful for *in vitro* studies. In addition we discuss the usefulness of our approach in more detail in *Conclusions*.

"Similar to fluorescence microscopy, the present approach can identify different polypeptides, but has two advantages: (i) it beats the diffraction limit and enables mapping with 20 nm spatial resolution, and (ii) it avoids the need to attach a bulky label for monitoring, which might affect the chemical properties and modify the interactions to be studied.

For very many applications, isotope labeling does not present an obstacle because it can be achieved either by expression of recombinant, labeled proteins or by chemical synthesis of labeled peptides. Studying the interaction of one particular polypeptide – e.g. A β – with several other partners can be done in an economic way as it requires only to label the common polypeptide of interest (A β), while each of the other interaction partners may remain unlabeled in binary mixtures. The approach can be extended to more than two interacting polypeptides by imaging one labeled interaction partner at a time in a series of multi-component mixtures.

Thus we expect that isotope-edition of nanoscale infrared images will be useful for studying interacting polypeptides in *in vitro* studies that serve to understand the natural complexity of biological processes. It will be particularly beneficial for amyloidogenic peptides/proteins where such interactions have medical relevance."

The authors propose using difference s-SNOM phase images between two wavelengths (1632 cm⁻¹, characteristic of unlabeled β -sheets and 1587 cm⁻¹, characteristic of ¹³C β -sheets) to locate the positions of labeled and unlabeled peptides mixed samples. I believe that this type of data analysis has not been reported previously for s-SNOM images but is quite similar to image ratios used for semiquantitative analysis in AFM-IR (another AFM-based nanoscale spectroscopy method that uses the AFM tip to transduce the sample thermal expansion). See for example Analyst, 2016, 141, 4848–4854. As the authors point out, such approach has limitations due to image drift that can even vary during a single image scan.

Thank you also for pointing us to this interesting publication, dealing with anomalous intensity effect in AFM-IR measurements and a way to deal with them by ratioing images taken at different wavenumbers. We agree that this image processing is similar to ours and we do not claim originality for our data evaluation. To make this clear, we use now the word "employed" instead of "introduced" when we first mention the difference images:

"To avoid any ambiguity, we employed an objective way to compare two phase images: we calculated difference images where the phase at 1629 cm⁻¹ was subtracted from the phase at 1587 cm⁻¹. "

Similar image processing approaches (subtraction, ratioing, overlay) have a long tradition in imaging, which we state now in the text:

"Such imaging of spectral properties at particular wavenumbers or wavelengths has been proven valuable in infrared and Raman microscopy to map particular functional groups and in fluorescence microscopy to map one or several specific molecules."

The manuscript is clearly written and while I appreciate the analytical effort put forward by the authors, unluckily I don't find the analysis very useful nor particularly reliable, due to the limitations highlighted by the authors with the analysis of control samples. An additional, important, concern is that while ^{13}C -labelling has the effect of shifting the amide I peak of unlabeled β -sheets from 1632 cm^{-1} to 1587 cm^{-1} , i.e. outside the amide I range for the other conformers (alpha helix etc.) the shifted peak for the isotopically labelled beta sheet (about 1587 cm^{-1}) is within the amide II frequency range of unlabeled peptides. Therefore, personally I would not rely on nor use such analysis for the study of peptides' conformations and, in my opinion, it will not be generally adopted by scientists in this field. Instead, I would trust the well vetted analysis provided by amide I (and possibly amide II) peak position deconvolution to identify the secondary structure of peptide and proteins, which is based on full spectra rather than 2 single frequencies. This approach is widely used in the analysis FTIR and AFM-IR spectra (see ref 26 and several others).

While the reviewer is critical to our approach, he/she does not point to a particular problem regarding the quality of our work. We were very honest in describing our controls and intentionally challenged the approach when we selected the lines along which the phase values are reported in Figs. S4 and S5. We regard this rather a strength of our manuscript than a weakness. The mentioned limitations all apply to some of the smallest aggregates, which are naturally most difficult to characterize. Here the approach (as all approaches) is limited by the signal to noise ratio of the current instruments but it is not limited in principle. Technical advances in the near future will increase the quality of the spectroscopic data, which then will be able to distinguish even very small particles. This aspect is now commented on in *Conclusions*:

"Ongoing technological developments are expected to push the detection limit to even smaller aggregates or molecules."

For all intermediate and large aggregates, but also for most small aggregates, the controls demonstrate the expected behavior.

The reviewer is concerned about applying our approach to β -sheets. For such secondary structures, the ^{13}C amide I band shifts toward the amide II band moving largely into the minimum between the amide I and II bands. This adds to the existing absorption in that region, which is however also affected by labeling (carboxylates and amide II band) so that both – the amide I and the amide II band – shift down considerably. We do not quite understand why this should be problematic for distinguishing between labeled and unlabeled peptides. The stronger absorption at 1587 cm^{-1} is only one effect of labeling. The other effect is the weaker absorption at 1629 cm^{-1} . The new Fig. S2 clearly shows that the absorption at 1587 cm^{-1} is stronger than that at 1629 cm^{-1} for the labeled peptide, whereas the opposite is true for the unlabeled peptide. Because we rely on both wavenumbers for isotope-editing of our images, we can safely distinguish between labeled and unlabeled peptides, which we also demonstrated in our experiments.

We find that the final comment is not related to our work because we did not intend to analyze the secondary structure of our peptides on the nanoscale. Instead, we demonstrate that we can distinguish different peptides by isotopic labeling even without resorting to full scale spectra. Having full spectra at each pixel is of course desirable, but hyperspectral imaging would take much longer than imaging at two particular wavenumbers and has several disadvantages. This aspect is now discussed in *Conclusions*.

"In future applications, full-range spectra at locations of interest identified by the isotope-edited images will be a valuable complement to the morphology information of the individual interaction partners. While it even would be desirable to have full spectral information for the entire area of nanoscale images (hyperspectral imaging), monitoring at a few selected wavenumbers dramatically speeds up the recording of the images and thus minimizes the risk of tip contamination or tip damage during the experiment. It

will enable researchers to scan larger sample areas than with hyperspectral imaging leading to a more representative sampling of the sample properties."

Combining full spectral deconvolution and isotopic labelling, would better achieve the manuscript goals but much more reliably so. Unluckily, state-of-the-art nanoFTIR s-SNOM is not great for recording full spectra because of the low spectral resolution (typically 16 cm⁻¹), long spectral acquisition times (typically 1 to 5 min) and generally low signal to noise. In contrast, AFM-IR spectra, has much higher spectral resolution (typically 1 cm⁻¹), shorter acquisition time (10 to 30 s) and appears better suited for nanoscale conformational analysis of peptides and proteins. Therefore, in my opinion, the manuscript doesn't provide significant advances to justify the publication in Communications Chemistry and may be better suitable in a very specialized journal (for example Analyst).

Our goal was not to record full spectra for a secondary structure analysis. Instead, the achievement of this work is to have identified the location of each of the mixture components by isotope-editing and two-wavenumber imaging. We agree regarding the resolution and would argue that standard approaches for secondary structure analysis need to be thought over and modified for s-SNOM and probably AFM-IR spectra for several reasons, one of which is the polarization of the tip-enhanced near-field which leads to dichroic effects for oriented samples. We discuss such effects now in text at the beginning of *Results and Discussion* when we describe the spectrum of Aβ40.

The reviewer is critical to our results because he/she judges the signal to noise ratio in s-SNOM spectra to be inferior to that of AFM-IR spectra. An advantage of the AFM-IR technique for spectrum recording is the use of an infrared laser that sweeps through the spectrum. This results in higher laser intensity per wavenumber interval than available with the broad band laser used to record s-SNOM spectra. A higher laser intensity improves the signal to noise ratio. However, the s-SNOM imaging mode also uses a monochromatic laser and has thus the same advantage over the s-SNOM spectrum mode as AFM-IR. The s-SNOM imaging and spectrum modes are essentially performed on two different instruments, that differ in light source, optical path, interferometer and detector. The only common parts are the AFM unit and the mirror that focuses the beam on the AFM tip. Therefore, one cannot easily judge the performance of one of them based on the performance of the other. Since the core of our results was obtained with the imaging mode, we find that noise in s-SNOM spectra cannot be used as an argument against our approach. We claim however, that labeled and unlabeled peptides can also be clearly distinguished in our spectra (see new Fig. S2).

We have considered buying an AFM-IR instrument, but the tests before our purchase were in favor of the s-SNOM instrument. In discussions with experienced AFM-IR users in protein science, I get the impression that small aggregates and thin fibers – as studied in our work – are a challenge also for the AFM-IR technique and that it sometimes can generate amide I signals that are hard to explain from the custom knowledge of protein FTIR spectra. The standard approach at present seems to use gold-coated tips. These contaminate easily when used for biological samples. If this happens during imaging, the experiment needs to be repeated. Therefore I am skeptical to the reviewer's opinion that imaging with full spectral information using AFM-IR should be considerably better than s-SNOM imaging at selected wavenumbers.

Few additional minor comments follow:

- On page 4 "topological" should be "topographic"

Thank you for pointing out this mistake, which has been corrected.

- Please specify in the SI the laser power and spectral range used to obtain the nanoFTIR spectra in fig 1.

We realized that our value for the band width in the SI was misleading since it referred to the total band width of the laser, which comprises five different ranges. One has to select a particular range for a given experiment. The spectral range of the broad band laser in our experiments was 1300 cm^{-1} to 2150 cm^{-1} . This range is now stated in the SI, as well as the laser power:

"The spectra were recorded at the second harmonic of the tip oscillation frequency using a broadband infrared laser ($1300\text{--}2150\text{ cm}^{-1}$, $0.7\text{--}0.8\text{ mW}$), 16 cm^{-1} spectral resolution, a zero-filling factor of 4, 10 ms integration time for each of the 2048 interferogram data points, and averaging 5 or 15 interferograms with the nano-FTIR tips or the ARRPW-NCPT tips, respectively. Recording of one spectrum took thus $\sim 100\text{ s}$ or $\sim 5\text{ min}$."

- The nanoFTIR spectroscopy experimental is not particularly clear. Please double check it. Why if the integration time is set to 10 ms and 5 to 15 interferograms were recorded the total a spectrum acquisition time is 1.5 to 5 min?

The integration time referred to each individual data point in the interferogram. This is now made clearer in the SI (see the text quoted for the above comment). This results in 20 s per interferogram, which gives a total measuring time of $\sim 100\text{ s}$ for 5 averaged interferograms and $\sim 5\text{ min}$ for 15 averaged interferograms. In addition, we now state that the zero-filling factor was 4.

Reviewers' comments:

Reviewer #1 (Remarks to the Author):

Manuscript untitled: « 13C-isotope-edting of nanoscale infrared mages reveals the action of an inhibitory peptide against amyloid- β aggregation » renamed “13C, 15N-labeling distinguishes different polypeptides in nanoscale images of the infrared absorption. Test case: interaction of the amyloid- β peptide with an inhibitory peptide.”

Thank you to the authors for all the efforts they put to answer my questions: Still, I have some comments.

Review 1:

1. The introduction should be reinforced, directing it more sincerely towards methodology, especially if this article is to be published in a journal like comm.chem. From my point of view, if the authors insist so much (especially in the title) on the application in biology, it would seem to me more judicious to publish this article in comm.biol. But again, not as it is.

Thank you for the answer, title, abstract and introduction are easier to read.

Review2: Some new comments for the introduction:

1.1. first line “Nanoscale infrared spectroscopy¹ is currently revolutionizing the spatial analysis of materials since it combines the morphological information provided by atomic force microscopy (AFM) with the chemical information from infrared spectroscopy.” Please be precise and define IR nanospectroscopy: report to the bibliography. Snom is not the only method to perform IR nanospectroscopy see a lot of references in the domain: list the different configurations of snom, (nanoFTIR, a-snom, TIR s-snom...) photothermal based techniques AFM-IR (contact, tapping...). => then justify rapidly the use of nanoFTIR. From this point, please avoid the use of the too-general “Nanoscale infrared spectroscopy” expression.

1.2. Do you have a reference showing that the 13C, 15N-labeling does not impact the secondary structure of a protein?

1.3. At the end of the introduction could you please present the outline of your article?

Review1:

2. Why did they choose this NCAM1-Prp hybrid peptide for interactions with A β 40-42? Even if it's clearly described on their previous paper, we need a short reminder. What puzzle me more is the following: some research groups have shown that the species bound to PrPC are predominantly small oligomer (not only fibrils). Would not it be the same for the hybrid peptide: as the author have the technology to study molecular interaction at the fibrillar level why not exploring this aspect?

ok

3. Furthermore, PrP 23–27 region is located within the part of the prion protein that is natively unstructured. If we look at their previous paper (ref 39), compared to A β 42, the IR absorption spectrum of this peptide does not exhibit a high absorption at 1629 cm⁻¹ but a broad band centered around 1645-1640 cm⁻¹? So why the authors only acquired the 1629 cm⁻¹ IR map? How the authors justify the lack of spectral data about this hybrid (I mean in the present paper)? This point is crucial for the consistency of

the paper. It's a serious methodological issue.

Sorry this point is still unclear. Once again, could you please add nanoFTIR spectra of pure NCAM1-Prp to your paper. Furthermore how is it possible that "A β 40 is partly unstructured"? May be I misunderstood something, so I might be wrong but when you read the literature it's not what is commonly known. So even if there are some tip effects (polarization of the near field along the tip axis...), that favored some molecular vibrations, the spectra of A β 40 fibril should exhibit at least shoulder at around 1628 cm⁻¹??

4. Then, spectral measurements were limited to bundles of fibers whereas we are waiting for measurements on individual fibers. Is there a reason? It's almost the same problem for imaging. The interpretation of the signal is more or less clear for bundles but not for singles fibers. The images should clarify this point but as there is a lot of bundles it's impossible to conclude even when looking carefully to supplementary figures.

Ok

5. From the spectra it seems that there are a lot of variability. How do the authors explain this observation? Why do we observe so many oscillations on spectra 1 (reference)? Why is the amide II band more intense than the amide I band for the unlabeled A β but not for the labeled A β ? Still we observe a lot of oscillations on the IR spectra, why?

The answer of the authors was the following:

We record the spectra at the low resolution of 16 cm⁻¹ in order to increase the signal to noise ratio. However, this implies only a few data points in the amide I region where the noise then can manifest as oscillation. We discuss the oscillations now in the text:

"We note that all spectra seem to contain oscillations, which is in contrast to the expected smooth appearance of the protein and the control spectra. We tentatively ascribe the "oscillations" to the noise. Noise leads to a deviation of the measured data points from the correct spectrum. Since the wavenumber interval between two data points is 16 cm⁻¹ at the spectral resolution used (ignoring the data points introduced by zero-filling), the noise will generate "ups and downs" in the measured spectrum with a spacing of 16 cm⁻¹. This gives the impression of an oscillation with a periodicity of 32 cm⁻¹, corresponding to twice the resolution. This is in fact the main "oscillation" period observed in the data."

In advance sorry if I'm a little bit rude: Since when a 16 cm⁻¹ resolution is the solution to solve S/N ratio? Furthermore, if you have such a resolution, please plot your spectra as a point spectrum and not as a line.

6. What kind of data processing was done on the spectra? It would be useful to have access to the raw spectra.

Ok

7. Figure 2, 3, 4: Why the authors do not show the optical amplitude at 1587 cm⁻¹?

Ok

8. Figure 4 top panel: the hybrid peptide seems to exhibit a low signal at 1629 cm⁻¹. Why the signal is higher when it's mixed with the labeled A β 40 ?

Ok

9. In the conclusion, the authors claim that "These observations indicate that NCAM1-PrP is able to coat or to dissolve existing A β 40 fibrils": I don't understand how the authors arrive to this conclusion just by looking at figure4 and S4.

Ok

10. In the SI: the part related to purple membrane is completely irrelevant. The authors should remove it.

Reviewer #2 (Remarks to the Author):

Dr Barth and his colleagues have carefully and convincingly answered questions raised by the reviewers. Their answers have been included and result in an almost complete rewriting of large parts of their manuscript. They made a great effort of clarifying most points raised and in improving their final text for future readers. I perfectly agree with these revisions.

Reviewer #3 (Remarks to the Author):

It was not my intention to upset the authors but to provide honest feedback. I stand by my comments that sample analysis using a broad spectral range is much more robust than analysis using 2 single wavelengths.

I believe that the authors improved the manuscript clarity upon revision. The authors also have better contextualized their research and the relevance and limitations of their method.

5/29/23

Answers to reviewer 1.

The text below contains only those comments of the reviewer that required an action from our side. The numbering refers to his/her original numbering. The changes in the text are highlighted in yellow in the manuscript file.

Review2: Some new comments for the introduction:

1.1. first line "Nanoscale infrared spectroscopy1 is currently revolutionizing the spatial analysis of materials since it combines the morphological information provided by atomic force microscopy (AFM) with the chemical information from infrared spectroscopy." Please be precise and define IR nanospectroscopy: report to the bibliography. Snom is not the only method to perform IR nanospectroscopy see a lot of references in the domain: list the different configurations of snom, (nanoFTIR, a-snom, TIR s-snom...) photothermal based techniques AFM-IR (contact, tapping...). => then justify rapidly the use of nanoFTIR. From this point, please avoid the use of the too-general "Nanoscale infrared spectroscopy" expression.

It was not our intention to use the term nanoscale IR spectroscopy as a synonym for the s-SNOM version of the technique. In contrast, we deliberately cited in the first sentence one of the few review articles that covers several of the implementations of this concept. We have now specified the different techniques at the beginning of Introduction without going too much into detail. The new text reads:

There are different technical realizations of nanoscale infrared spectroscopy, which all probe the interaction of a sample with infrared light, but analyze different types of sample responses: photothermal near-field imaging (PTIR or AFM-IR),³ photo-induced force microscopy (PiFM),^{4,5} and scanning near-field optical microscopy (s-SNOM).⁶⁻⁸ The latter is used in this work and has two operation modes: one for spectrum recording with a broad band infrared laser and one for imaging at a particular wavenumber using a monochromatic laser. These approaches are termed here nano-Fourier transform infrared (FTIR) spectroscopy and s-SNOM infrared imaging, respectively.

While the different techniques have their specific problems and benefits, they provide similar information. Therefore general terms like "nanoscale infrared spectroscopy" have their justification. For example, the method-oriented part of *Introduction* is relevant for all nanoscale methods and the cited articles have used AFM-IR not s-SNOM. We would also claim that our results and our method are relevant for all nanoscale IR techniques, not only for s-SNOM. This is now mentioned at the beginning of *Conclusions*:

"While we demonstrated its [refers to isotope-edited nanoscale imaging] feasibility for s-SNOM infrared imaging, the concept is also applicable for other nanoscale vibrational spectroscopy methods, like PTIR (AFM-IR), PiFM, and tip-enhanced Raman spectroscopy."

Department of Biochemistry and Biophysics

Stockholm University
Department of Biochemistry
and Biophysics
Svante Arrhenius väg 16C
SE-106 91 Stockholm, Sweden

Visiting address:
Svante Arrhenius väg 16C
Room K311
<https://www.su.se/english/profiles/abart>

Phone: +46-8-162452
Telefax: +46-8-155597
E-mail: barth@dbb.su.se

Therefore, we would like to keep the general term in *Introduction* and *Conclusion*. We have however, taken care to specify our method in *Results and Discussion*, in particular in the section titles, where we use the terms *nano-FTIR* for spectra and *s-SNOM infrared imaging* for the images.

Our choice of s-SNOM was guided by instrument tests in the context of the purchase of our instrument where an AFM-IR instrument did not perform well. However, there are high-quality AFM-IR data in the literature, so we do not believe that s-SNOM is generally better than AFM-IR, but neither that it is worse (see answer to reviewer 3 in the last round). Our reason for using s-SNOM is its availability of this technique in our instrument and we therefore prefer not to discuss in the manuscript why we used s-SNOM.

1.2. Do you have a reference showing that the ^{13}C , ^{15}N -labeling does not impact the secondary structure of a protein?

^{13}C - and ^{15}N - labeling is standard for structure determination of proteins by NMR spectroscopy (now mentioned in *Introduction*) and according to the NMR experts in our department, it is generally accepted that the isotopes do not modify the structure. Such changes would be easy to detect by an isotope effect on the proton chemical shift, which is a sensitive indicator of protein secondary structure. For a given C-H group, one chemical shift signal is observed for the ^{12}C -isotope. This signal is split into two for the ^{13}C -isotope. The average ^{13}C -H chemical shift is the same as the ^{12}C -H chemical shift, if the secondary structure is the same. We are not aware of reports that contradict this expectation. A ^{13}C - or ^{15}N -isotope effect on the secondary structure is certainly not debated in the NMR community since it is surprisingly difficult to find scientific studies on this topic. We found articles that support (Pudney et al. 2013, JACS 135, 2512-2517) and challenge (Ranasinghe et al. 2019, J. Phys. Chem. B 123, 10403–10409; Hartmann et al. 2003, BioMetals 16, 379-382) the general view. Nevertheless, if labeling caused significant structural changes, these should have been revealed in comparisons between NMR and X-ray structures. But this is not the case: such comparisons demonstrate strong similarities between structures obtained with the two methods (Billeter 1992, Quart. Rev. Biophys. 25, 325-377, Garbuzynskiy et al. 2005, Proteins 60, 139-147). The observed differences between the atomic positions obtained by the two methods are on the level of differences between the different NMR structures in the structure ensemble of a given pdb file (Sikic et al. 2010, Open Biochem. J. 4, 83-95) or somewhat larger (Everett et al. 2016, Protein Sci. 25, 30-45), but they reduce when modern NMR methods like restrained Rosetta refinement are used (Everett et al. 2016, Protein Sci. 25, 30-45). They are particularly low for the conformation of β -strands (Sikic et al. 2010, Open Biochem. J. 4, 83-95), which is the secondary structure of interest in our work.

The structural differences between NMR and X-ray structures are explained by method-specific assumptions for the evaluation of the raw data, different data treatments, and by the different conditions in solution or in a crystal (Billeter 1992, Quart. Rev. Biophys. 25, 325-377, Garbuzynskiy et al. 2005, Proteins 60, 139-147, Sikic et al. 2010, Open Biochem. J. 4, 83-95). None of the articles discusses isotope labeling as the reason for the observed differences. We note also that the level of structural information from the infrared absorption spectrum is by far less detailed than the atomic information from the NMR studies. Thus, even if the small discrepancies between NMR and X-ray structures were caused by isotope labeling, we expect that the isotope effects on the infrared spectrum would be caused by the mass effect and not by a modified structure.

1.3. At the end of the introduction could you please present the outline of your article?

The new text at the end of Introduction reads:

"This work reports s-SNOM measurements in the infrared spectral range of ^{13}C , ^{15}N -labeled A β 40 and of unlabeled A β 40, A β 42, and NCAM1-PrP, either in pure form or in mixtures. We discuss first the nano-

FTIR spectra of A β 40 and its s-SNOM infrared images recorded at two different wavenumbers of which one detects mainly the absorbance of unlabeled peptide and the other mainly the absorbance of labeled peptide. The images of labeled A β 40 and unlabeled A β 40 and of their mixture serve to verify the concept of isotope-edited imaging. As a further proof of concept, we analyze images of A β 42 protofibrils and their mixture with labeled A β 40 fibrils, which demonstrate that the peptide identity determined from the isotope-edited infrared image correlates with the expected aggregate type. Finally, we discuss nano-FTIR spectra and images of NCAM1-PrP fibrils and the images of their mixture with labeled A β 40 fibrils, which reveal the anti-amyloid effect of the former peptide."

3. Furthermore, PrP 23–27 region is located within the part of the prion protein that is natively unstructured. If we look at their previous paper (ref 39), compared to A β 42, the IR absorption spectrum of this peptide does not exhibit a high absorption at 1629 cm⁻¹ but a broad band centered around 1645-1640 cm⁻¹? So why the authors only acquired the 1629 cm⁻¹ IR map? How the authors justify the lack of spectral data about this hybrid (I mean in the present paper)? This point is crucial for the consistency of the paper. It's a serious methodological issue.

Sorry this point is still unclear. Once again, could you please add nanoFTIR spectra of pure NCAM1-Prp to your paper. Furthermore how is it possible that "A β 40 is partly unstructured"? May be I misunderstood something, so I might be wrong but when you read the literature it's not what is commonly known. So even if there are some tip effects (polarization of the near field along the tip axis...), that favored some molecular vibrations, the spectra of A β 40 fibril should exhibit at least shoulder at around 1628 cm⁻¹??

We did not add nano-FTIR spectrum of pure NCAM1-PrP fibrils to the previous version because we thought that the clarification of our mistake in the Król et al. article would resolve the issue. We have now recorded the requested spectrum, added it to Fig. S2 and discussed it in the text as follows:

"Nano-FTIR spectrum and s-SNOM infrared imaging of NCAM1-PrP. The nano-FTIR spectrum of NCAM1-PrP is shown in Fig. S2. It exhibits amide I and II maxima at 1640 and 1533 cm⁻¹, respectively, which both indicate β -sheet structure. The low amide II band position indicates additionally an antiparallel arrangement of the β -strands in contrast to the parallel orientation in A β 40 fibrils discussed above. Antiparallel β -sheets of the NCAM1-PrP fibrils are also indicated by their FTIR spectrum (dried from H₂O solution, measured by attenuated total reflection, not shown), which exhibits typical antiparallel β -sheet features: a distinct high wavenumber amide I band at 1696 cm⁻¹, a main amide I band at 1627 cm⁻¹, and an amide II maximum at 1526 cm⁻¹. While the amide II band positions are close in nano-FTIR and FTIR spectra, the amide I band positions are significantly different, which we ascribe to the polarization effect discussed above. A strong low wavenumber β -sheet signal would require the β -sheets to be oriented perpendicular to the substrate surface with strands running parallel to the surface. The former is not possible given the thinness of the fibrils (see below).

In spite of the differences between the nano-FTIR spectrum and the FTIR spectrum in the amide I range, the nano-FTIR spectrum demonstrates that the signal at 1629 cm⁻¹ is stronger than that at 1587 cm⁻¹ and therefore that the wavenumbers used for imaging in the previous sections are also suitable for studying mixtures of unlabeled NCAM1-PrP and labeled A β 40."

Regarding the choice of the higher wavenumber for imaging (1629 cm⁻¹), we note additionally that we were limited by the wavenumber range of the laser used. The intensity of this laser drops steeply from 1560 to 1650 cm⁻¹, where it is close to zero. Thus, even if the nano-FTIR maximum is at 1640 cm⁻¹, using this wavenumber for imaging would not have improved the signal to noise ratio due to the smaller laser intensity.

The mentioned height of the NCAM1-PrP fibrils is now analyzed in the subsequent paragraph:

"Individual fibrils have a height of 2.7 nm (average from 40 fibrils, standard deviation 0.2 nm) while the thickest aggregates are ~5 nm high."

Regarding the question of "partly unstructured A β 40", I was referring to the Król et al. article, in particular to Fig. 3 in that article, which shows the infrared spectra. When the peptide is properly dissolved, it is largely unstructured at first and aggregates over the time course of several hours upon which it forms β -sheets. The lack of structure at the beginning of aggregation is shown by its CD spectrum in Fig. 2 and the infrared spectrum in Fig. 3 of the Król et al. article. The infrared spectrum indicates already the additional presence of some β -sheets, since aggregation is faster at the higher peptide concentration used for the infrared experiment. Later stages of the aggregation process were not monitored by infrared spectroscopy in this article. We did this, however, in earlier work (Abelein et al., JACS 2016, Doi: 10.1021/jacs.6b04511) where a strong β -sheet band near 1620 cm^{-1} can be observed in D₂O. D₂O was used in that work because the spectrum in solution was of interest, whereas the present work used H₂O because the solvent does not matter spectroscopically as it needs to be removed for the nano-FTIR measurements. Note that band positions in D₂O are lower than in H₂O. In summary, the infrared spectrum of A β 40 in the Król et al. article does not correspond to the spectrum expected for A β 40 fibrils studied in this work. The nano-FTIR spectra of the A β 40 fibrils shown in Fig. 1 of the submitted manuscript have their maximum near 1630 cm^{-1} as expected for β -sheets and as already described in the text.

5. ...In advance sorry If I'm a little bit rude: Since when a 16 cm-1 resolution is the solution to solve S/N ratio? Furthermore, if you have such a resolution, please plot your spectra as a point spectrum and not as a line.

That a lower resolution reduces the perceived noise is a known effect. This holds for nano-FTIR spectra, conventional FTIR spectra and even for dispersive instruments. The former two use both an interferometer and generate spectra by Fourier transformation of the interferogram. In an interferogram, "sharp" features of a spectrum like noise or bands of rotational vibrational transitions of gases are encoded far away from the white light position (where the path difference between the two beams in the interferogram is zero and all wavelengths interfere constructively). This region is not recorded when a low spectral resolution is selected, and consequently sharp features do not appear in low resolution spectra. The figures below show this effect in practice.

Fig. 1. Spectra recorded at different resolutions with an FTIR instrument equipped with a diamond ATR unit and a DTGS detector. The absorbance was obtained from two consecutive scans without sample. The resolutions were 1 cm^{-1} (blue), 2 cm^{-1} (red), 4 cm^{-1} (green), 8 cm^{-1} (turquoise), and 16 cm^{-1} (orange). Spectra were vertically shifted for a clearer presentation. The spectra show increased noise in regions of low light intensity and increased noise at higher resolutions.

Fig. 2. Nano-FTIR spectra of the purple membrane obtained at different resolutions: 16.7 cm^{-1} (purple), 12 cm^{-1} (turquoise), 10 cm^{-1} (green), and 8 cm^{-1} (red).

We prefer not to follow the recommendation of the reviewer to plot the spectra as point spectra for the following reasons:

1. We are not aware of published point spectra even when the settings are similar to ours and therefore rather stick to the common convention of plotting line spectra.
2. The spectral data points in our spectra have a spacing of 4 cm^{-1} , although the resolution is 16 cm^{-1} . This is because the instrument uses a zerofilling factor of 4 to generate the spectra. This data point spacing is close to the conventional spacing of 2 cm^{-1} in FTIR spectra recorded at 4 cm^{-1} resolution with a zerofilling factor of 2 and does not motivate a different way of plotting the data.
3. Plotting points instead of lines would make the figures very confusing.

We hope that the additional explanations and measurements dissipate the remaining concerns of reviewer 1.

Yours sincerely

Professor Andreas Barth

REVIEWERS' COMMENTS:

Reviewer #1 (Remarks to the Author):

Dear colleagues,

Many thanks for patiently answering all my questions. From my point of view, the manuscript is clearer and gains in quality (if I might express it like this). I have also learned a lot. So I'm really grateful.

I still can't entirely agree with some experimental choices but It's how science goes.

To conclude, I have no more concerns. I agree with the revisions.